# Less Is More in Federated Continual Learning: RieSelect for Conflict-Aware Layer Selection in LLMs

**Wenqi Qiu** [1]   **Yipeng Zhou** [2]   **Lin Zhu** [3]   **Laizhong Cui** [1]

## Abstract

Federated continual learning (FCL) of large language models on edge devices is constrained by a communication–stability–plasticity trilemma. We reveal a less-is-more phenomenon: beyond a moderate layer upload ratio, stability loss offsets saturated plasticity gains, so overall continual performance no longer improves. Moreover, layer-wise conflict is heavy-tailed and concentrates in a few layers; denser uplink increasingly includes these layers, which disproportionately drive forgetting and motivates selective sparse communication. Therefore, we introduce RieSelect, which treats stability as staying within a Fisher-metric safe basin around historical solutions. Under this safe-basin constraint, we derive a layer-wise conflict score and a closed-form certified safe step size for finite local updates, and formulate selective uplink as a knapsack-based utility–risk selection, balancing plasticity gains against stability risks. Extensive experiments show that, under a per-round uplink budget, RieSelect achieves the best performance across task orders. Beyond this matched-budget setting, under standard communication protocols, RieSelect improves average accuracy by 18.99–28.14 points while reducing total uplink by 53–115$\times$.

## 1. Introduction

The deployment landscape of Large Language Models (LLMs) is undergoing a paradigm shift from centralized pre-training to decentralized adaptation on edge devices (Zheng

[1]College of Computer Science and Software Engineering, Shenzhen University, Shenzhen, China. [2]School of Computing, Faculty of Science and Engineering, Macquarie University, Sydney, Australia. [3]China Mobile Jiutian Artificial Intelligence Technology (Beijing) Co., Ltd. Correspondence to: Laizhong Cui <cuilz@szu.edu.cn>.

*Proceedings of the 43rd International Conference on Machine Learning*, Seoul, South Korea. PMLR 306, 2026. Copyright 2026 by the author(s).

et al., 2025; Wang et al., 2025). Unlike static cloud environments, these edge scenarios are inherently dynamic, where heterogeneous data emerges as continuous, non-stationary streams rather than fixed batches (Wang et al., 2024b; Li et al., 2025b). To maintain relevance across such evolving contexts without compromising user privacy, Federated Continual Learning (FCL) has emerged as a critical paradigm (Ma et al., 2022; Yang et al., 2024). It enables the global model to incrementally assimilate skills from these local task streams, collaboratively evolving over time without centralizing raw user data.

Enabling billion-parameter LLMs for lifelong FCL exposes an under-explored trilemma among communication efficiency, stability, and plasticity (Wang et al., 2024b; Hamedi et al., 2025). Even with Parameter-Efficient Fine-Tuning (PEFT) that freezes the backbone (Houlsby et al., 2019; Lester et al., 2021; Hu et al., 2022), sequential adaptation still requires frequent uplink transmissions, which are costly on bandwidth-constrained edges. For billion-parameter LLMs, even a tiny trainable fraction can translate to billion-scale updates per round. Meanwhile, non-i.i.d. local streams further exacerbate the stability–plasticity tension (Yoon et al., 2021; Dong et al., 2022; Yu et al., 2024). Critically, uplink is not just a resource constraint but can also shape continual dynamics non-trivially. We therefore focus on the uplink bottleneck and formalize communication efficiency as continual adaptation under a per-round uplink budget. This poses a central challenge: can FCL jointly achieve communication efficiency, stability, and plasticity?

Under such budgets, existing approaches remain limited in jointly balancing stability and plasticity. In PEFT-based FCL, methods typically choose between adapter isolation (Wang et al., 2023; Yang et al., 2025a) and indiscriminate sharing (Hu et al., 2022; Smith et al., 2023), trading stability for transfer or sacrificing stability for plasticity. Meanwhile, classical safeguards focus on either parameter importance (Zenke et al., 2017; Chaudhry et al., 2018) or update direction (Lopez-Paz & Ranzato, 2017; Chaudhry et al., 2019a) without pinpointing sparse high-conflict components. Moreover, federated extensions often add replay (Chaudhry et al., 2019b) or dense auxiliary statistics (Salami et al., 2025; Guo et al., 2024) that can overwhelm the uplink bud-

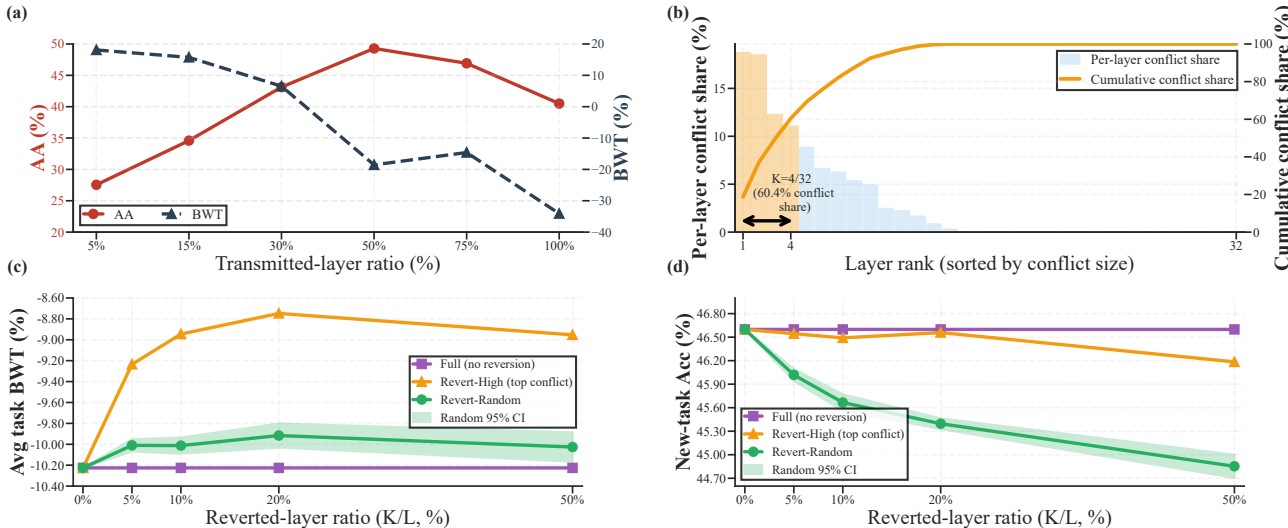

*Figure 1.* **Less-is-more arises from concentrated cross-task conflict.** (a) Average accuracy (AA; left) and backward transfer (BWT; right; more negative indicates more forgetting) versus the transmitted-layer ratio. For ratios $< 100\%$, we use a budget-matched Top-$K$ selection based on the $\ell_2$ update norm. (b) Layer-wise cross-task gradient conflict under full uplink, measured by $\max(0, -\cos(g_{\text{curr}}, g_{\text{past}}))$ computed with replayed past gradients for diagnosis. (c) Averaged task BWT versus the reverted-layer ratio $K/L$ when reverting $K$ uplinked layers, compared with full uplink. (d) Plasticity (measured by New-task accuracy) versus $K/L$ under the same settings.

get and erode PEFT's efficiency gains.

To address this trilemma, we challenge the common assumption that more uplink communication monotonically improves federated adaptation (McMahan et al., 2017; Li et al., 2020). Fig. 1(a) reveals a less-is-more regime: beyond a moderate layer upload ratio, overall continual performance ceases to improve. Additional uplink increasingly harms stability, offsetting saturated plasticity gains. This non-monotonicity suggests that uplink can carry both useful and conflicting updates.

To pinpoint where the conflict comes from, we diagnose cross-task gradient conflict under full uplink as the layer-wise cosine opposition between the current update direction and an aggregated historical-retention proxy (Lopez-Paz & Ranzato, 2017; Chaudhry et al., 2019a). We observe a heavy-tailed pattern: a few layers dominate the total conflict (Fig. 1(b)). This suggests a simple mechanism for Fig. 1(a): as the transmitted-layer ratio increases, uplink becomes more likely to include these high-conflict layers, which disproportionately overwrite previously consolidated behaviors. Consistent with this, a targeted reversion ablation shows that reverting only the most conflicting layers markedly improves stability, with little impact on plasticity (Fig. 1(c-d)). Together, these results indicate that forgetting under dense uplink is driven by a small set of high-conflict layers. Effective communication should therefore target them. Additional settings and results are in Appendix A.

This calls for deciding which layer updates can be communicated without destabilizing past behaviors. We adopt a Riemannian view where the Fisher Information Matrix (FIM)

defines the local KL geometry and induces an anisotropic safe basin around historical solutions, constraining updates by predictive KL drift. From this basin view, we propose **RieSelect**. It scores layer-wise conflict in the Fisher metric for the current update, caps step sizes on risky layers with a closed-form certificate to stay within the safe basin, and selects layers via a 0–1 knapsack to maximize gain under the per-round uplink budget.

Our experiments on sequential NLP benchmarks with T5-Large and LLaMA2-7B show that, under a fixed per-round uplink budget, RieSelect achieves the best average accuracy across task orders. Beyond the matched-budget setting, when comparing against existing methods under their standard communication protocols, RieSelect improves AA by up to 28.14 and 18.99 points while reducing total uplink traffic by up to $53\times$ on T5-Large and $115\times$ on LLaMA2-7B.

## 2. Related Work

**Continual Learning**. Prior CL methods mainly fall into three families: regularization, gradient projection, and replay (Wang et al., 2024a; Shi et al., 2025). Regularization-based approaches such as EWC (Kirkpatrick et al., 2017) and its variants (Zenke et al., 2017; Chaudhry et al., 2018) leverage Fisher information to estimate parameter importance and penalize changes to important weights, but typically do not explicitly reason about update-direction conflicts. Projection-based methods including GEM (Lopez-Paz & Ranzato, 2017) and A-GEM (Chaudhry et al., 2019a) enforce inequality constraints on gradients to avoid negative transfer. Replay methods (Chaudhry et al., 2019b; Riemer

et al., 2019) mitigate forgetting by revisiting past examples, but are hard to deploy in federated settings due to privacy and data-retention constraints.

**Continual PEFT**. To tailor LLMs to downstream tasks efficiently, PEFT has emerged as a dominant paradigm (Ding et al., 2023; Han et al., 2024; Zhang et al., 2026). Low-Rank Adaptation (LoRA) (Hu et al., 2022; Dettmers et al., 2023) is widely adopted for its balance of performance and efficiency (Biderman et al., 2024; Shuttleworth et al., 2026; Zeng & Lee, 2024). Extending PEFT to continual settings, recent works integrate LoRA with CL mechanisms (Zhou et al., 2024; Yang et al., 2025b). These approaches largely fall into two structural choices: C-LoRA (Smith et al., 2023) applies self-regularization on shared low-rank parameters, while O-LoRA (Wang et al., 2023), N-LoRA (Yang et al., 2025a), and HydraLoRA (Tian et al., 2024) enforce task-specific isolation via new adapters or experts. Overall, existing continual LoRA methods mainly control forgetting through adapter-level sharing or isolation, leaving limited room for finer-grained coordination within a shared update.

**Federated Continual Learning**. FCL aims to mitigate catastrophic forgetting under federated settings (Wang et al., 2024b; Yang et al., 2024; Hamedi et al., 2025). Early works typically combine FedAvg with local CL mechanisms (Hendryx et al., 2021; Yoon et al., 2021; Zhang et al., 2023; Dong et al., 2022), and recent frameworks further integrate PEFT to shrink transmitted updates (Lu et al., 2023; Guo et al., 2023; Zhang et al., 2024). Nevertheless, several FCL methods still rely on additional uplink signals beyond PEFT updates for stable consolidation, such as uploading class prototypes and feature summaries in PILoRA (Guo et al., 2024) or transmitting per-layer Gram statistics for closed-form merging in LoRM (Salami et al., 2025). When models are large, these auxiliary payloads can dominate the uplink budget and dilute the efficiency gains of PEFT, leaving robust FCL under tight communication constraints challenging.

## 3. Preliminaries

We consider a federated learning system with $N$ clients collaboratively training a global model $\Theta \in \mathbb{R}^D$ by minimizing $\min_\Theta \sum_{k=1}^N p_k \mathcal{L}_k(\Theta)$, where $\sum_{k=1}^N p_k = 1$ and $\mathcal{L}_k$ is the local objective on client $k$. Training proceeds in communication rounds $m = 0, 1, \ldots, M$. At round $m$, the server broadcasts $\Theta^{(m)}$ to a subset of clients $\mathcal{S}_m$, each of which runs $E$ steps of local optimization starting from $\Theta^{(m)}$ to obtain an updated model $\Theta_k^{(m)}$. The server aggregates the local models as $\Theta^{(m+1)} \leftarrow \sum_{k \in \mathcal{S}_m} p_k \Theta_k^{(m)}$.

We extend this setting to FCL, where each client observes a non-stationary data stream indexed by tasks $t \in \{1, \ldots, T\}$. At task $t$, client $k$ only accesses its current data $\mathcal{D}_{k,t}$ and does not store past data $\mathcal{D}_{k,<t} := \bigcup_{\tau=1}^{t-1} \mathcal{D}_{k,\tau}$. We define the

task loss as $\mathcal{L}_t(\Theta) := \sum_{k=1}^N p_k \mathcal{L}_{k,t}(\Theta; \mathcal{D}_{k,t})$ and the historical loss as $\mathcal{L}_{<t}(\Theta) := \sum_{\tau=1}^{t-1} \mathcal{L}_\tau(\Theta)$. Since $\mathcal{L}_{<t}(\Theta)$ is intractable to evaluate under no replay, we cast FCL as minimizing $\mathcal{L}_t$ under a bounded historical-loss increase (Lopez-Paz & Ranzato, 2017; Chaudhry et al., 2019a):

$$\min_\Theta \ \mathcal{L}_t(\Theta) \quad \text{s.t.} \quad \mathcal{L}_{<t}(\Theta) \leq \mathcal{L}_{<t}(\bar{\Theta}) + \epsilon, \qquad (1)$$

where $\bar{\Theta}$ denotes the parameters obtained after training task $t-1$ and $\epsilon \geq 0$ specifies the tolerated increase.

Federated adaptation of LLMs is communication intensive, as synchronizing full model parameters across clients is prohibitive (Yoon et al., 2021; Qiu et al., 2025). We adopt PEFT and restrict optimization to low-rank updates via LoRA. Given a frozen pre-trained weight matrix $W_0 \in \mathbb{R}^{d_{\text{out}} \times d_{\text{in}}}$, LoRA introduces $A \in \mathbb{R}^{r \times d_{\text{in}}}$ and $B \in \mathbb{R}^{d_{\text{out}} \times r}$ with rank $r \ll \min(d_{\text{out}}, d_{\text{in}})$ and parameterizes the update as $\Delta W = BA$, yielding $h = W_0 x + \Delta W x$. We denote the trainable low-rank parameters as $\phi = \{A, B\}$ and keep $W_0$ frozen. In FCL, only $\phi$ is synchronized and aggregated across clients. We further write the vectorized LoRA parameters as $\theta = \text{vec}(\phi) \in \mathbb{R}^d$, and use $\bar{\theta}$ to denote the historical reference point obtained after task $t - 1$, maintained via the online Fisher summaries in Sec. 4.2.

Under no replay, the historical loss $\mathcal{L}_{<t}(\theta)$ in Eq. (1) is unavailable, so we approximate it around $\bar{\theta}$ with a quadratic surrogate: $\mathcal{L}_{<t}(\theta) \approx \mathcal{L}_{<t}(\bar{\theta}) + \frac{1}{2}(\theta - \bar{\theta})^\top \mathbf{H}_{<t}(\theta - \bar{\theta})$. For probabilistic models, the Fisher information $F(\bar{\theta}) = \mathbb{E}\left[\nabla_\theta \log p(y|x; \bar{\theta}) \nabla_\theta \log p(y|x; \bar{\theta})^\top\right]$ approximates the local KL metric (Kullback & Leibler, 1951; Amari, 1998). For efficiency (Kirkpatrick et al., 2017; Chaudhry et al., 2018), we approximate $\mathbf{H}_{<t}$ with a diagonal Fisher surrogate $\bar{F}$, updated online at task boundaries using only current-task data (see Sec. 4.2), which yields the replay-free displacement $r^2(\theta) = \|\theta - \bar{\theta}\|_{\bar{F}}^2$ and constraint $\frac{1}{2}r^2(\theta) \leq \epsilon$.

## 4. Method

### 4.1. Overview: A Geometric Framework

Eq. (1) constrains degradation on previous tasks, but it is intractable under no replay since $\mathcal{L}_{<t}$ cannot be evaluated. As established in the preliminaries, we rely on the Fisher-weighted quadratic proxy $r^2(\theta)$, which measures displacement under the local metric induced by the FIM. However, federated updates are noisy and non-infinitesimal: on a curved parameter manifold, a finite step that appears small in this local quadratic proxy can correspond to a much larger intrinsic deviation from the historical region, creating a curvature trap. This motivates a Riemannian treatment of stability, where historical knowledge is protected by a Fisher safe basin and the step size is chosen in a curvature-aware manner.

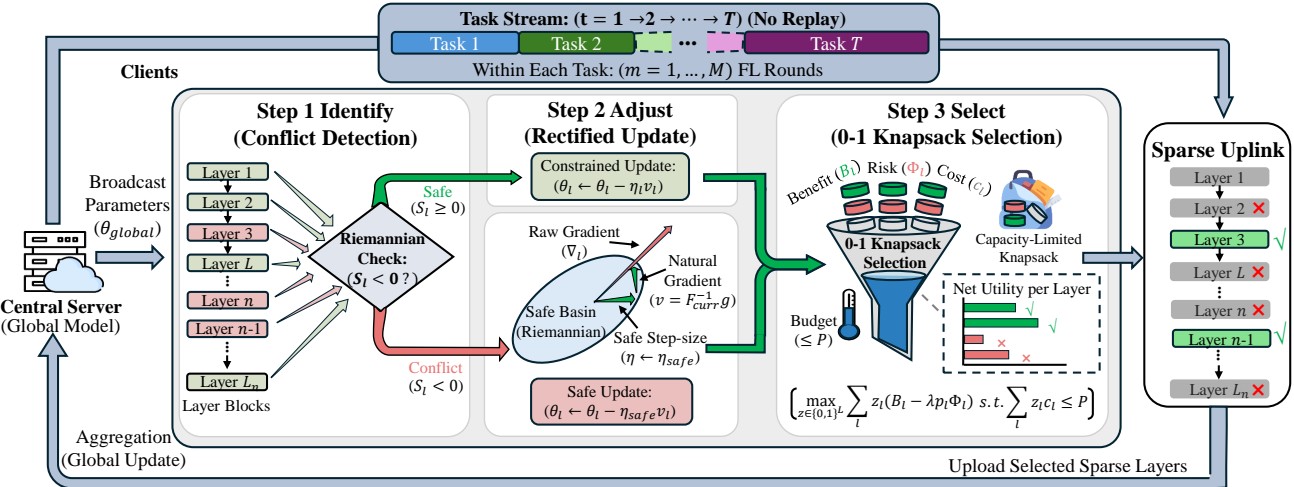

*Figure 2.* **Overview of RieSelect.** A three-step workflow. **Identify** (Sec. 4.2): each client computes a layer conflict score $S_\ell$ from replay-free Fisher statistics. **Adjust** (Sec. 4.3): layers with $S_\ell < 0$ (red) are step-size capped by the closed-form certificate $\eta_{\text{safe}}$ (Eq. 8) to stay within the safe basin, while other layers (green) use the lightweight basin cap (Eq. 10). **Select** (Sec. 4.4): given per-layer utility $B_\ell$ and risk $\Phi_\ell$, the client solves a 0–1 knapsack under budget $P$ to decide which layers to transmit.

Guided by this safe-basin view, RieSelect decomposes into three layer-wise subroutines (Fig. 2). **Identify** (Sec. 4.2): each client computes a layer conflict score $S_\ell$ quantifying how the current layer update tends to move the model out of the Fisher-metric safe basin formed by past tasks. **Adjust** (Sec. 4.3): when $S_\ell < 0$, we apply a closed-form certified safe step size $\eta_{\text{safe}}$ so the local update remains within the safe basin. **Select** (Sec. 4.4): using these per-layer risk/utility estimates, we select layers to upload under the per-round uplink budget by solving a 0–1 knapsack.

### 4.2. Identify: Conflict Detection

We model each past task $j < t$ by a replay-free local quadratic surrogate in the LoRA parameter space, $\mathcal{L}_j(\theta) \approx \mathcal{L}_j(\theta_j^*) + \frac{1}{2}(\theta - \theta_j^*)^\top F_j(\theta - \theta_j^*)$, where $\theta_j^*$ is the (approximate) stationary point after task $j$ and $F_j$ is a diagonal Fisher surrogate (Kirkpatrick et al., 2017; Chaudhry et al., 2018). Aggregating all past tasks yields a compact, replay-free closed-form expression for the historical proxy gradient. Specifically, let $\bar{F} := \sum_{j<t} w_j F_j$ and $\bar{z} := \sum_{j<t} w_j (F_j \odot \theta_j^*)$. Then $\bar{\theta} := \bar{z} \oslash \bar{F}$, and $\nabla_\theta \mathcal{L}_{<t}(\theta) \approx \bar{F} \odot (\theta - \bar{\theta})$. This summarizes all past tasks using only $(\bar{F}, \bar{z})$, avoiding explicit storage of $\{(\theta_j^*, F_j)\}_{j<t}$.

Each client maintains $(\bar{F}, \bar{z})$ locally and updates them once per task without transmitting auxiliary statistics to the server. The safe-basin constraint is enforced client-wise and the server only aggregates already-constrained updates. Let $F_t$ be the diagonal Fisher surrogate estimated from current-task data at the end-of-task iterate $\theta_t^*$. We perform a task-wise exponential moving average $\bar{F} \leftarrow \gamma \bar{F} + (1-\gamma) F_t$ and $\bar{z} \leftarrow \gamma \bar{z} + (1-\gamma)(F_t \odot \theta_t^*)$, which immediately updates $\bar{\theta} \leftarrow \bar{z} \oslash \bar{F}$. This task-wise update maintains a streaming Fisher

summary across tasks using only current-task data (Schwarz et al., 2018; Lee et al., 2025).

To control forgetting under Eq. (1), we use the Fisher-weighted quadratic proxy $r^2(\theta) = \|\theta - \bar{\theta}\|_{\bar{F}}^2$ (equivalently $r^2(\theta) \leq 2\epsilon$). Let $g$ and $F_{\text{curr}}$ denote the current-task gradient and its diagonal Fisher surrogate, and define the natural-gradient direction $v := F_{\text{curr}}^{-1} g$. For the update $\theta^+ = \theta - \eta v$, a second-order expansion of the historical proxy yields

$$\begin{aligned}
\Delta \mathcal{L}_{<t} &:= \mathcal{L}_{<t}(\theta^+) - \mathcal{L}_{<t}(\theta) \\
&\approx -\eta\, v^\top \nabla_\theta \mathcal{L}_{<t}(\theta) + \frac{\eta^2}{2}\, v^\top \bar{F} v,
\end{aligned} \quad (2)$$

where $\nabla_\theta \mathcal{L}_{<t}(\theta)$ is the replay-free gradient proxy from above. With diagonal $\bar{F}$, the historical increment in Eq. (2) decomposes additively across layers. Restricting the update $\theta^+ = \theta - \eta v$ to layer $\ell$ gives

$$\Delta \mathcal{L}_{<t}^{(\ell)} \approx -\eta\, S_\ell + \frac{\eta^2}{2}\, Q_\ell, \quad (3)$$

where $S_\ell$ is defined in Eq. (4) and $Q_\ell := \sum_{i \in \ell} v_i \bar{F}_i v_i \geq 0$. This yields a sharp conflict criterion: if $S_\ell < 0$, then $\Delta \mathcal{L}_{<t}^{(\ell)} > 0$ for any $\eta > 0$. We therefore mark layers with $S_\ell < 0$ as conflicting and use this as the first stage of our workflow. Moreover, the conflict signal is actionable. Under the same diagonal Fisher proxy, attenuating the update on layers with $S_\ell < 0$ provably decreases the historical-loss increment. The proof is provided in Appendix B.1.

In FCL, local gradients and update directions are noisy due to mini-batch stochasticity and data heterogeneity, making coordinate-wise filtering statistically brittle. We therefore score conflict at layer granularity. Recall that $v := F_{\text{curr}}^{-1} g$ is the Fisher-preconditioned update direction. The key

quantity governing its first-order interaction with the historical proxy is the inner product $v^\top \nabla_\theta \mathcal{L}_{<t}(\theta)$. Using $\nabla_\theta \mathcal{L}_{<t}(\theta) \approx \bar{F} \odot (\theta - \bar{\theta})$, this interaction decomposes as $v^\top \nabla_\theta \mathcal{L}_{<t}(\theta) = \sum_i v_i \bar{F}_i (\theta_i - \bar{\theta}_i)$. We thus define the coordinate interaction score $s_i := v_i \bar{F}_i (\theta_i - \bar{\theta}_i)$ and aggregate it within each LoRA layer $\ell$ (index set $i \in \ell$, size $n_\ell := |\ell|$) to obtain

$$S_\ell := \sum_{i \in \ell} s_i = \sum_{i \in \ell} v_i \bar{F}_i (\theta_i - \bar{\theta}_i), \qquad (4)$$

which corresponds to the linear term in the layer-wise increment of the historical proxy in Eq. (3). Moreover, under zero-mean noise with bounded within-layer correlation, the normalized variance satisfies $\mathrm{Var}(S_\ell/n_\ell) \leq \sigma^2_{\max,\ell}/n_{\mathrm{eff},\ell}$, where $n_{\mathrm{eff},\ell} = n_\ell/[1 + (n_\ell - 1)\rho_{\max,\ell}]$. The proof and a sign-misclassification bound are provided in Appendix B.2. Beyond stability, this granularity reduces the candidate space from $d$ coordinates to $L$ blocks, so subsequent selection scales with $L \ll d$.

### 4.3. Adjust: Rectified Update

The above analysis yields a sharp directional conflict signal via the layer score $S_\ell$. However, directional screening alone is insufficient under finite and multi-step updates. While $S_\ell$ characterizes first-order alignment in the local tangent space, feasibility is governed by the intrinsic drift from the historical reference under the Fisher metric. On a curved manifold, the coordinate update $\theta^+ = \theta - \eta v$ can deviate from the geodesic displacement, so the quadratic proxy $r^2(\theta)$ can misestimate the basin under non-infinitesimal steps. Therefore, for layers flagged as conflicting, we further control the update magnitude in a second stage.

We define the set of conflicting layers as $\mathcal{I} := \{\ell : S_\ell < 0\}$, then consider a particular layer $\ell \in \mathcal{I}$ and formalize magnitude control over its LoRA parameter manifold. To simplify notation, we restrict all quantities to this layer and drop the layer subscript in what follows. Let $d_F(\cdot, \cdot)$ denote the geodesic distance induced by the Fisher information metric. Motivated by the proxy constraint $r^2(\theta) \leq 2\epsilon$, we define the Riemannian safe basin around $\bar{\theta}$ as

$$\mathcal{B}_\epsilon(\bar{\theta}) := \{\theta : d_F^2(\theta, \bar{\theta}) \leq 2\epsilon\}. \qquad (5)$$

Because the Fisher metric captures local changes in model predictions, $d_F$ measures distributional drift rather than coordinate drift. We therefore choose the update magnitude so that the post-update iterate stays in $\mathcal{B}_\epsilon(\bar{\theta})$ on layers in $\mathcal{I}$.

The basin in Eq. (5) is intrinsic but not directly computable, since evaluating $d_F(\theta, \bar{\theta})$ requires solving a geodesic problem under the Fisher metric. A common surrogate replaces $d_F^2(\theta, \bar{\theta})$ with the quadratic proxy $r^2(\theta)$, which matches the second-order expansion of the KL divergence at $\bar{\theta}$. For

finite steps, however, curvature introduces a higher-order mismatch, quantified below.

**Lemma 4.1** (Geodesic–quadratic mismatch). *Assume the manifold has bounded sectional curvature in a normal neighborhood $\mathcal{N}$ of $\bar{\theta}$, with lower/upper bounds $-\kappa$ and $\mathcal{K}$, and let $\kappa_{\max} := \max\{\kappa, \mathcal{K}\}$. Then there exists a normal neighborhood $\mathcal{N}_0 \subseteq \mathcal{N}$ of $\bar{\theta}$ and a constant $C > 0$ such that for any $\theta \in \mathcal{N}_0$,*

$$d_F^2(\theta, \bar{\theta}) = \Delta\theta^\top \bar{F} \Delta\theta + \mathcal{E}(\theta), \qquad (6)$$

*where $\Delta\theta := \theta - \bar{\theta}$ and $|\mathcal{E}(\theta)| \leq C \kappa_{\max} \|\bar{F}\| \|\Delta\theta\|^4$.*

The proof is provided in Appendix B.3. Lemma 4.1 implies that $r^2(\theta)$ is only locally accurate. Thus, enforcing $r^2(\theta) \leq 2\epsilon$ does not necessarily guarantee $\theta \in \mathcal{B}_\epsilon(\bar{\theta})$ under finite updates. We therefore derive a computable sufficient condition by upper bounding $d_F^2(\theta^+, \bar{\theta})$ along the update, which will yield a certified safe step size.

To obtain a computable certificate for the intrinsic constraint in Eq. (5), we invoke Riemannian comparison results that relate the geodesic distance to its tangent-space quadratic form under bounded curvature.

**Theorem 4.2** (Comparison bounds for geodesic distance). *Assume bounded sectional curvature in a normal neighborhood $\mathcal{N}$ of $\bar{\theta}$, with lower/upper bounds $-\kappa$ and $\mathcal{K}$. Let $R_\epsilon := \sqrt{2\epsilon}$ and, when $\mathcal{K} > 0$, assume $R_\epsilon < \pi/(2\sqrt{\mathcal{K}})$. Then for any $\theta \in \mathcal{N}$ with $d_F(\theta, \bar{\theta}) \leq R_\epsilon$,*

$$\psi_{+\mathcal{K}}(R_\epsilon) \, r^2(\theta) \leq d_F^2(\theta, \bar{\theta}) \leq \psi_{-\kappa}(R_\epsilon) \, r^2(\theta), \quad (7)$$

*where $\psi_\pm$ are curvature comparison factors. Moreover, as $R_\epsilon \to 0$, $\psi_{+\mathcal{K}}(R_\epsilon) = 1 - \frac{\mathcal{K}}{3} R_\epsilon^2 + O(\kappa_{\max} R_\epsilon^2) + O(\mathcal{K}^2 R_\epsilon^4)$, and $\psi_{-\kappa}(R_\epsilon) = 1 + \frac{\kappa}{3} R_\epsilon^2 + O(\kappa_{\max} R_\epsilon^2) + O(\kappa^2 R_\epsilon^4)$.*

The proof of Theorem 4.2 is deferred to Appendix B.4. We use its upper bound to enforce the sufficient certificate $\beta_\epsilon r^2(\theta^+) \leq R_\epsilon^2$, where $\beta_\epsilon$ denotes any valid upper comparison constant. In particular, Theorem 4.2 provides the (generally non-computable) choice $\beta_\epsilon = \psi_{-\kappa}(R_\epsilon)$ when a curvature lower bound $\kappa$ is available, and we derive a closed-form maximal step size along the layer update $\theta^+ = \theta - \eta v$.

**Proposition 4.3** (Maximal safe step size for $\ell \in \mathcal{I}$). *Fix a conflicting layer $\ell \in \mathcal{I}$ and consider all quantities restricted to this layer. Let $\vartheta, \varsigma \geq 0$ be margins accounting for estimation errors. Define the residual $\delta_\epsilon := R_\epsilon^2 - \beta_\epsilon(r^2(\theta^+) + \varsigma)$. If $\delta_\epsilon \geq 0$ and $Q_\ell > 0$, the maximal certified step size that guarantees $d_F^2(\theta^+, \bar{\theta}) \leq R_\epsilon^2$ under the certificate $\beta_\epsilon r^2(\theta^+) \leq R_\epsilon^2$ is*

$$\eta_{safe} = \frac{1}{Q_\ell} \left[ (S_\ell - \vartheta) + \sqrt{(S_\ell - \vartheta)^2 + \frac{Q_\ell}{\beta_\epsilon} \delta_\epsilon} \right]. \quad (8)$$

*If $\delta_\epsilon < 0$ or $Q_\ell = 0$, no positive step can be certified and we set $\eta_{safe} = 0$.*

The proof is deferred to Appendix B.5. Proposition 4.3 yields a closed-form controller whose only non-observable term is $\beta_\epsilon$. Rather than estimating $\kappa$, we interpret $\beta_\epsilon$ as a metric inflation factor that conservatively upper-bounds the intrinsic-drift stretch of the current Fisher geometry relative to the historical metric $\bar{F}$ within the safe region. Under our diagonal Fisher approximation, this inflation admits a layer-wise generalized-eigenvalue characterization and reduces to the maximum coordinate-wise Fisher ratio. We take $F_{\text{curr}}$ to be a diagonal envelope such that $F(\theta) \preceq F_{\text{curr}}$ holds for all $\theta$ in the certified neighborhood. Consequently, we instantiate $\beta_\epsilon$ with the computable proxy

$$\hat{\beta}_\ell := \max_{i \in \ell} \frac{\max\{\bar{F}_i, F_{\text{curr},i}\}}{\bar{F}_i}, \tag{9}$$

and substitute $\hat{\beta}_\ell$ for $\beta_\epsilon$ in Eq. (8). Since the certified step size is non-increasing in $\beta_\epsilon$, this plug-in choice yields a conservative controller. The detailed proof is provided in Appendix B.6.

Finally, even for non-conflicting layers $\ell \notin \mathcal{I}$, a finite step can still increase the quadratic proxy through the second-order term. Since $S_\ell \geq 0$ implies $r^2(\theta^+) = r^2(\theta) - 2\eta S_\ell + \eta^2 Q_\ell \leq r^2(\theta) + \eta^2 Q_\ell$, we conservatively clip the step size to keep the proxy within the basin radius:

$$\eta_\ell = \min\Big\{\eta_0, \sqrt{\frac{R_\epsilon^2/\hat{\beta}_\ell - r_\ell^2(\theta)}{Q_\ell}}\Big\}, \tag{10}$$

and set $\eta_\ell = 0$ when the square root is not real. This cap is used whenever $\ell \notin \mathcal{I}$ and requires only replay-free scalars.

### 4.4. Select: 0–1 Knapsack

Certified step-size control rectifies high-conflict layer updates, but uplink remains the bottleneck in FCL for LLMs. We therefore transmit only a subset of LoRA layers, prioritizing high learning utility while penalizing persistent conflict and endpoint forgetting.

To keep selection replay-free, each layer $\ell$ is summarized by scalars computed from the same $E$ local steps, using layer-wise step sizes $\eta_{\ell,e}$ ($\eta_{\text{safe}}$ for $\ell \in \mathcal{I}$, and the conservative cap in Eq. (10) otherwise). At step $e$, we derive the per-step benefit and forgetting proxies $B_{\ell,e} := [(\eta_{\ell,e} - \frac{1}{2}\eta_{\ell,e}^2) g_e^{(\ell)\top} v_e^{(\ell)}]_+$ and $\Delta\Phi_{\ell,e} := -\eta_{\ell,e} S_{\ell,e} + \frac{1}{2}\eta_{\ell,e}^2 Q_{\ell,e}$, where $[\cdot]_+ := \max\{\cdot, 0\}$ and $Q_{\ell,e}$ is the step-$e$ instantiation of $Q_\ell$ in Eq. (3). Intuitively, $B_{\ell,e}$ measures "how much we gain now," while $\Delta\Phi_{\ell,e}$ measures "how much we may forget," both using only replay-free local statistics from the same $E$ steps. Moreover, using $r_{\ell,e+1}^2 - r_{\ell,e}^2 = -2\eta_{\ell,e} S_{\ell,e} + \eta_{\ell,e}^2 Q_{\ell,e}$, the forgetting proxy telescopes over $E$ steps and admits the endpoint form (equivalently, the clipped sum of $\Delta\Phi_{\ell,e}$ over $e = 0, \ldots, E-1$): $\Phi_{\ell,m} := \left[\frac{1}{2}(r_{\ell,E}^2 - r_{\ell,0}^2)\right]_+$.

Derivations of these proxies and the telescoping relation are provided in Appendix B.7.

In communication round $m$, we aggregate benefits $B_{\ell,m} := \sum_{e=0}^{E-1} B_{\ell,e}$ and measure how persistently a layer conflicts via $p_{\ell,m} := \frac{1}{E} \sum_{e=0}^{E-1} \mathbb{I}[S_{\ell,e} < 0]$. Let $c_\ell$ denote the uplink cost of layer $\ell$, and let $P$ be a constant per-round uplink traffic budget. We select a subset of layers $Z_m \subseteq \{1, \ldots, L\}$ to upload by solving

$$\max_{Z_m} \sum_{\ell \in Z_m} \Big(B_{\ell,m} - \lambda\, p_{\ell,m} \Phi_{\ell,m}\Big) \quad \text{s.t.} \sum_{\ell \in Z_m} c_\ell \leq P. \tag{11}$$

This favors layers with high utility while penalizing large endpoint forgetting when conflict is frequent. Since each layer is either uploaded or not, Eq. (11) is a 0–1 knapsack problem. We solve it via standard dynamic programming (Bellman, 1966) in $O(LP)$ time.

Overall, the per-round overhead is $O(d)$, since all geometric quantities are computed by element-wise operations with diagonal Fisher surrogates. The subset selection adds $O(LP)$ time via dynamic programming, which is negligible in our setting with $L \ll d$ and a fixed per-round uplink budget $P$. Appendix C reports empirical runtime overhead measurements supporting this claim.

## 5. Experiments

### 5.1. Experimental Setup

**Architectures and Datasets.** We conduct experiments using two distinct backbones, T5-Large (Chung et al., 2024) and LLaMA2-7B (Touvron et al., 2023), to evaluate efficacy across encoder-decoder and decoder-only architectures. Following established protocols (Zhao et al., 2024; Liang et al., 2026; Li et al., 2025a), we evaluate on the SuperNI (Wang et al., 2022) and Long Sequence (Razdaibiedina et al., 2023) benchmarks. SuperNI comprises 15 diverse NLP tasks including generation and extraction organized into two sequences denoted as Order 1 and Order 2, while Long Sequence consists of 15 classification tasks forming Order 3 and Order 4. For comprehensive details on task composition and sequences, please refer to Appendix D.1.

**Compared Baselines.** We compare our method against three representative paradigms: (a) **PEFT-based CL**: This category includes LoRA (Hu et al., 2022), O-LoRA (Wang et al., 2023), N-LoRA (Yang et al., 2025a), HydraLoRA (Tian et al., 2024), and C-LoRA (Smith et al., 2023). These baselines represent structural strategies that balance between rigid parameter isolation and indiscriminate sharing. (b) **Traditional CL**: We adapt EWC (Kirkpatrick et al., 2017), A-GEM (Chaudhry et al., 2019a), and Replay (Chaudhry et al., 2019b) to the FL setting. These methods mitigate forgetting by leveraging regularization

*Table 1.* **Performance on LLaMA2-7B.** AA (%) and BWT (%) on SuperNI and Long Sequence. **Comm.** denotes the total *uplink* volume (GB) over all tasks, including LoRA updates and method-specific auxiliary statistics, such as Fisher matrices or prototype metadata. **Bold** and underlined numbers indicate the best and second-best results. Improvement reports gains over the best non-ours baseline.

| Type | Method | Comm. (GB) | SuperNI | | | | LongSeq | | | |
| --- | --- | --- | --- | --- | --- | --- | --- | --- | --- | --- |
| | | | Order 1 | | Order 2 | | Order 3 | | Order 4 | |
| | | | AA↑ | BWT↑ | AA↑ | BWT↑ | AA↑ | BWT↑ | AA↑ | BWT↑ |
| PEFT-CL | LoRA (Hu et al., 2022) | 2.93 | 32.87 | −11.24 | 31.14 | −15.96 | 40.50 | −34.10 | 43.83 | −26.72 |
| | O-LoRA (Wang et al., 2023) | 2.93 | 35.22 | −8.80 | 37.43 | 0.50 | 38.19 | −16.03 | 35.47 | −14.47 |
| | N-LoRA (Yang et al., 2025a) | 2.93 | 37.03 | −5.18 | 35.22 | 0.85 | 40.44 | −20.86 | 32.22 | −10.54 |
| | HydraLoRA (Tian et al., 2024) | 8.06 | 27.67 | −16.02 | 36.96 | −5.78 | 40.64 | −34.30 | 44.94 | −24.98 |
| | C-LoRA (Smith et al., 2023) | 2.93 | 34.30 | −9.67 | 38.11 | −2.05 | 50.77 | −16.97 | 48.17 | −17.92 |
| CL | EWC (Kirkpatrick et al., 2017) | 5.86 | 37.60 | −2.84 | 37.94 | 0.31 | 37.77 | −15.73 | 50.16 | −12.54 |
| | Replay (Chaudhry et al., 2019b) | 2.93 | 38.56 | **0.12** | 37.67 | −5.14 | 44.48 | -7.81 | 41.88 | -7.99 |
| | A-GEM (Chaudhry et al., 2019a) | 2.93 | 31.95 | −12.65 | 40.92 | −1.61 | 60.38 | −13.03 | 55.43 | −15.4 |
| FCL | LoRM (Salami et al., 2025) | 4.69 | 31.54 | −13.94 | 36.82 | −1.50 | 50.63 | −19.66 | 43.64 | −20.74 |
| | PILoRA (Guo et al., 2024) | 5.86 | 24.66 | −18.88 | 37.51 | −5.41 | 39.03 | −37.4 | 30.32 | −44.29 |
| Ours | **RieSelect** | **0.07** | **41.12** | -2.25 | **44.46** | **5.08** | **73.39** | **6.13** | **74.42** | **6.72** |
| | Improvement | ↓ 97.5% | +2.56 | -2.37 | +3.54 | +4.23 | +13.01 | +13.94 | +18.99 | +14.71 |

metrics or retaining historical exemplars. (c) **FCL Methods**: We evaluate PILoRA (Guo et al., 2024) and LoRM (Salami et al., 2025), representing state-of-the-art FCL solutions that utilize prototypes or module merging.

**Evaluation Metric.** We employ Average Accuracy (AA) to measure the overall performance on all learned tasks, and Backward Transfer (BWT) to quantify the forgetting of knowledge acquired from previous tasks. Let $A_{t,j}$ denote the test score on task $j$ after the model has finished training on task $t$, and $T$ be the total number of tasks. These metrics are defined as AA $= \frac{1}{T} \sum_{j=1}^{T} A_{T,j}$ and BWT $= \frac{1}{T-1} \sum_{j=1}^{T-1} (A_{T,j} - A_{j,j})$.

**Implementation Details.** We simulate a federated environment with 50 clients, sampling 5 clients per global round. To ensure a fair comparison, we apply LoRA adapters with a consistent rank of $r = 8$ across all methods. We impose a per-client per-round uplink traffic budget $P$ for RieSelect, measured by the payload size of transmitted LoRA updates. Each LoRA tensor $\ell$ has communication cost $c_\ell$ proportional to its update size, and we upload only a subset satisfying $\sum_{\ell \in Z_m} c_\ell \leq P$. Since the selected subset varies across rounds, we report uplink volume rather than a fixed layer count; replay-free Fisher summaries are maintained locally and are not uploaded. More detailed hyperparameters and budget settings are provided in Appendix D.2.

**5.2. Main Results**

Table 1 and Appendix Table 6 (T5-Large; Appendix D.3) report AA/BWT together with the resulting total uplink volume under each method's standard communication protocol. To further test backbone scalability, we additionally evaluate

*Table 2.* **Effectiveness of each step on LLaMA2-7B.** AA and BWT are reported on Order 2 and Order 3. $\Delta$ denotes Ours minus the ablated variant, shown as $\Delta$AA/ $\Delta$BWT.

| Variant | Order 2 | | | Order 3 | | |
| --- | --- | --- | --- | --- | --- | --- |
| | AA↑ | BWT↑ | $\Delta$AA/$\Delta$BWT | AA↑ | BWT↑ | $\Delta$AA/$\Delta$BWT |
| w/o Step 2 | 32.98 | −15.44 | +11.48/+20.52 | 35.23 | −38.18 | +38.16/+44.31 |
| w/o Step 3 | 34.46 | 2.32 | +10.00/+2.76 | 39.96 | 3.79 | +33.43/+2.34 |
| Ours | **44.46** | **5.08** | – | **73.39** | **6.13** | – |

RieSelect on Qwen2.5-14B-Instruct, with results reported in Appendix D.4. Note that only RieSelect enforces a fixed per-round uplink budget; budget-matched comparisons are deferred to Sec. 5.3. We provide three observations:

**Identifying and controlling conflict-heavy layers is key.** Most PEFT-CL and classical CL baselines fall short because they do not explicitly isolate these conflict-heavy components, making it hard to reduce forgetting without harming adaptation. Dense-sharing methods increasingly upload high-conflict layers and accumulate forgetting, whereas isolation-heavy methods over-protect previous tasks and reduce plasticity. In contrast, RieSelect achieves the highest AA across task orders, improving over the best non-ours baseline by up to **18.99** points on LLaMA2-7B and **28.14** on T5-Large. This advantage is enabled by geometry-aware conflict identification and safe-basin rectification, which mitigates destructive drift while preserving plasticity.

**Stability should be assessed together with plasticity.** Appendix Table 6 indicates that some methods can appear more stable by becoming overly conservative, thereby underfitting new tasks and failing to achieve the desired trade-off. Fig. 3 visualizes this effect on T5 Order 4 by tracking current-task accuracy and $AA_t$ (AA over tasks 1:$t$ evaluated after finish-

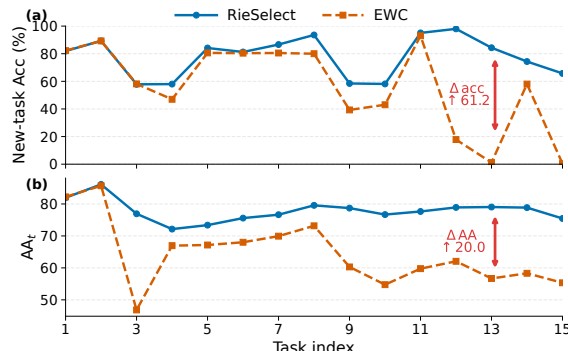

*Figure 3.* **Performance on T5-Large Order 4.** We track (a) current-task accuracy and (b) $AA_t$ over the task index for RieSelect and EWC. EWC progressively underfits later tasks, leading to a widening gap to RieSelect. Red arrows summarize the average gaps over Tasks 12–15 ($\Delta$acc ↑61.2, $\Delta$AA ↑20.0).

ing task $t$). EWC gradually loses new-task learning ability and therefore lags behind RieSelect throughout the sequence. In contrast, RieSelect constrains only high-conflict layers within the Riemannian safe basin, filtering destructive conflict without stalling useful learning.

**Less communication can be more effective.**

Table 1 and Appendix Table 6 show a wide traffic gap, yet more communication does not necessarily yield stronger continual adaptation. RieSelect achieves the best AA across orders while drastically reducing total traffic, requiring up to **53×** less uplink on T5-Large and up to **115×** less on LLaMA2-7B than the most communication-intensive baseline. This suggests that indiscriminate full transmission can amplify conflict by uploading high-conflict components, and auxiliary payloads further inflate uplink without guaranteed stability or plasticity gains. In contrast, RieSelect turns communication into a selective filter by uploading only a small set of high-utility and low-risk updates, preserving performance while sharply reducing traffic.

### 5.3. Selection, Not Sparsity, Drives Gains

Tables 1 and Appendix Table 6 show that strong performance can coexist with much less uplink, but this could be due to sparsity itself or to how sparsity is allocated. To separate these factors, we enforce the same per-round uplink budget for baselines by applying a budget-matched Top-K layer upload, while keeping the rest of the training protocol unchanged. Under this matched budget, baselines exhibit a clear AA drop in Fig. 4(a). Their BWT can appear less negative because later tasks are underfit rather than genuinely learned, which is consistent with the near-zero new-task accuracy in Fig. 4(b). In contrast, RieSelect remains strong under the same budget. Random selection and the results for Order 2 further corroborate this finding; see Appendix D.6.

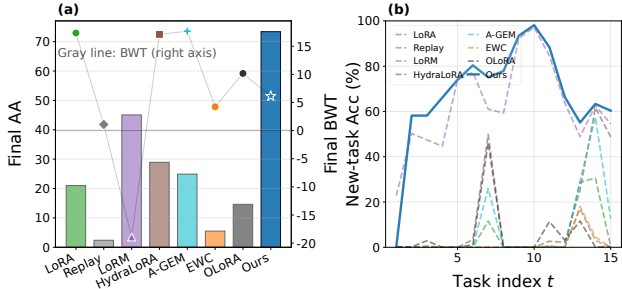

*Figure 4.* **Budget-matched Top-K layer uploads.** We match the per-round uplink budget $P$ for each baseline by transmitting only a Top-K subset of layers, ranked by the $\ell_2$ norm of layer-wise updates and selected greedily until the budget is met. **(a)** Final AA (left axis) and final BWT (right axis). **(b)** New-task accuracy along the task sequence, showing that Top-K baselines may underfit later tasks and thus make BWT appear less negative.

### 5.4. Ablation and Robustness Study

**Effectiveness of each step.** Table 2 shows that under matched budgets, the full method achieves the best AA and the strongest positive BWT across evaluated orders. When only the knapsack selection is kept, communication targets high-utility layers but conflict is uncorrected, so learning new tasks induces pronounced forgetting. When only the Riemannian adjustment is kept, forgetting is mitigated, yet random layer uploads dilute the scarce budget over low-utility layers and limit learning, resulting in weaker AA. Overall, Step 2 secures a stability floor and Step 3 lifts a plasticity ceiling, making them complementary and jointly necessary.

**Effectiveness of Hyperparameter $R_\epsilon$.** Fig. 5 sweeps $R_\epsilon$ on LLaMA2 Order 2 and reveals a clear sweet spot. With $R_\epsilon = 10$, RieSelect achieves the best final AA and is the only evaluated setting with positive final BWT. Smaller values of $R_\epsilon$ make the safe basin overly restrictive and suppress useful adaptation, whereas larger $R_\epsilon$ relax the constraint and allow conflict to accumulate, leading to degraded AA and more negative BWT. Overall, $R_\epsilon$ provides a direct stability–plasticity knob with a robust intermediate optimum.

**Robustness to stronger heterogeneity.** Appendix D.5 reports additional results under more skewed Dirichlet splits with $\alpha \in \{1, 0.5, 0.1\}$. RieSelect consistently achieves the highest AA and maintains positive BWT, indicating that its replay-free conflict estimates remain effective under stronger client heterogeneity. This suggests that RieSelect is robust not only to task-order variation but also to stronger client-level distribution shifts.

## 6. Conclusion

We study communication-efficient federated continual learning for large language models and show that uploading more updates can sometimes hurt continual performance. We in-

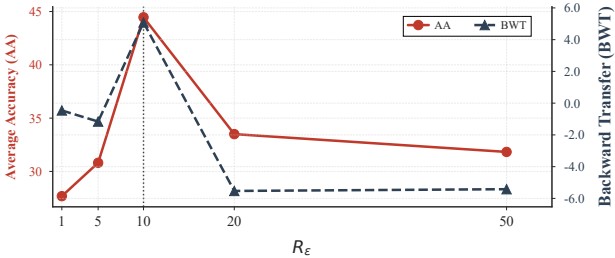

*Figure 5.* $R_\epsilon$ **sensitivity on LLaMA2-7B Order 2.** Final AA and final BWT as a function of $R_\epsilon$ under the same setting and uplink budget. An intermediate $R_\epsilon$ achieves the best trade-off, while overly small or large $R_\epsilon$ degrades both adaptation and retention.

troduce RieSelect to retain previously learned knowledge while enabling effective adaptation under a strict uplink budget. Our main insight is that cross-task conflict is concentrated in a small subset of layers, making selective transmission more suitable than dense communication. Guided by this insight, RieSelect identifies and rectifies high-risk updates within a Fisher-metric safe basin using a certified step-size cap, and selects layers to upload via a utility–risk knapsack formulation. Empirical results validate that selective, conflict-aware communication is an effective direction for uplink-constrained federated continual adaptation.

## 7. Limitations and Future Work

This work focuses on replay-free federated continual learning with PEFT-based transformer LLMs, and broader validation beyond this setting remains future work. RieSelect currently uses a fixed safe-basin radius $R_\epsilon$ across task transitions, while adaptive radius selection may further improve robustness. The method also requires local storage for Fisher summaries and introduces lightweight client-side computation for conflict estimation and layer selection, motivating future studies on lower-overhead deployment in more resource-constrained environments.

## Acknowledgements

This work has been partially supported by National Natural Science Foundation of China under Grant No. U23B2026 and No.62372305, Guangdong Basic and Applied Basic Research Foundation under Grant No. 2024B1515040012, Shenzhen Science and Technology Program under Grant No. KJZD20230923114809020 and Research Team Cultivation Program of Shenzhen University under Grant No.2023QNT015, Yongjiang Talent Program No. 2024A-392-G, Key Programs of Ningbo Municipal Natural Science Foundation No. 2024J021, and Key Project of the Zhejiang Provincial Natural Science Foundation No. LZ26F020010.

## Impact Statement

This paper presents work whose goal is to advance the field of Machine Learning. There are many potential societal consequences of our work, none which we feel must be specifically highlighted here.

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

# A. Additional Settings and Results for Fig. 1

### A.1. Experimental Settings

We conduct our experiments using the vanilla LoRA on LLaMA2-7B under Order 3, with a learning rate of $1 \times 10^{-5}$ and a total of 50 simulated clients. For each task, we run 5 global rounds and randomly select 5 clients per round. Each selected client performs 10 local training epochs. The heterogeneity of client data is governed by a Dirichlet parameter $\alpha = 10$. To lower memory usage, we use bf16 precision and gradient checkpointing. For details, please refer to Appendix D.

Unless otherwise stated, the sparse uplink results in Fig. 1(a) are obtained using a *budget-matched Top-K layer upload*: we rank each layer's transmitted update by the $\ell_2$ norm of its layer-wise update and greedily select layers until the per-round uplink budget $P$ is met (same protocol as the Top-$K$ baseline). We emphasize that the points with $< 100\%$ communication in Fig. 1(a) are obtained via a budget-matched Top-$K$ layer upload under the per-round budget $P$. We also experimented with *uniformly random* layer selection under the same budget $P$ (i.e., randomly picking layers to upload while respecting the same budget constraint), and observed the same qualitative trend as Top-$K$. This suggests that the phenomenon in Fig. 1(a) is not an artifact of the specific layer-selection heuristic.

### A.2. Plasticity Saturates Under Moderate Transmission

A natural alternative explanation for sparse communication is that it simply underfits new tasks and appears better only because it preserves previous ones. To rule this out, we measure *plasticity* as the test accuracy of each task immediately after it is learned, under different transmitted-layer ratios. Fig. 6 reports the mean plasticity across tasks. The shaded area denotes the 95% confidence interval across tasks in the sequence, computed as $\mu \pm t_{0.975,T-1} \cdot s/\sqrt{T}$ where $T$ is the number of tasks. We observe that plasticity quickly saturates under moderate transmission (e.g., 50%), and increasing transmission towards 100% yields only marginal gains within uncertainty, supporting a diminishing-return regime for new-task utility.

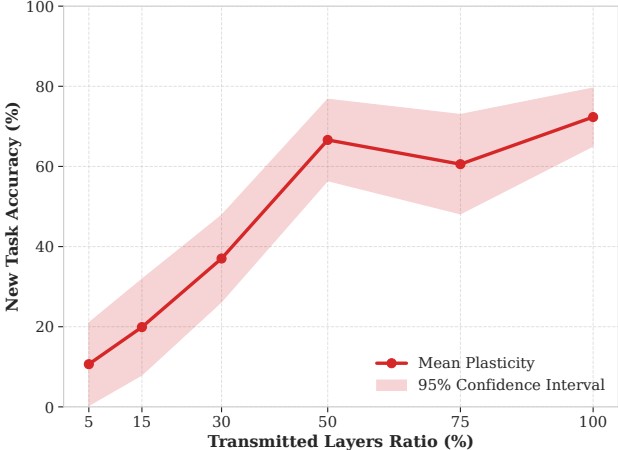

*Figure 6.* **Plasticity vs. transmission ratio.** Mean plasticity (new-task accuracy) under different transmitted-layer ratios.

### A.3. Layer-wise Cross-task Gradient Conflict and Its Concentration

This section explains how we produce Fig. 1(b) and the cross-boundary concentration curve reported in the appendix. At each task boundary $t$ (after finishing task $t-1$ and before adapting to task $t$), we measure how much the *current-task update direction* conflicts with a proxy for *historical retention*. The replay buffer $\mathcal{D}_{\text{past}}^{(t)}$ is used only for this diagnostic metric (Fig. 1(b)) and the concentration analysis. Our method is replay-free during training and instead uses the Fisher-based historical-retention proxy in Sec. 4.2.

We construct two probe sets: a small batch $\mathcal{D}_{\text{cur}}^{(t)}$ from the incoming task to represent plasticity, and a *past buffer* $\mathcal{D}_{\text{past}}^{(t)}$ sampled from a cumulative replay buffer that aggregates examples from previously seen tasks. The past buffer therefore represents an *all-previous-tasks* mixture rather than only the immediately preceding task.

We focus on the LoRA parameters and group them by transformer layer (one group per layer, indexed by $\ell \in \{1, \ldots, L\}$).

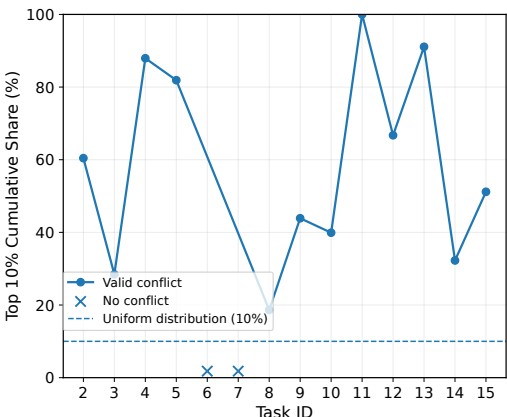

*Figure 7.* **Top 10% cumulative conflict share across task boundaries.** For each boundary $t$, we compute $\mathrm{Top10Share}(t)$ from the layer-wise conflict shares. The dashed line marks the $10\%$ uniform baseline; points substantially above it indicate that a small subset of layers consistently dominates total conflict. Markers labeled as "No conflict" correspond to boundaries where $\sum_\ell c_\ell(t) = 0$.

For each layer group, we compute the mean gradient on the two probes:

$$\mathbf{g}_{\mathrm{cur},\ell}^{(t)} = \frac{1}{|\mathcal{D}_{\mathrm{cur}}^{(t)}|} \sum_{(x,y)\in\mathcal{D}_{\mathrm{cur}}^{(t)}} \nabla_{\theta_\ell}\mathcal{L}(x,y), \tag{12}$$

$$\mathbf{g}_{\mathrm{past},\ell}^{(t)} = \frac{1}{|\mathcal{D}_{\mathrm{past}}^{(t)}|} \sum_{(x,y)\in\mathcal{D}_{\mathrm{past}}^{(t)}} \nabla_{\theta_\ell}\mathcal{L}(x,y). \tag{13}$$

We define layer-wise cross-task gradient conflict via cosine similarity and count only opposing directions as conflict:

$$\cos^{(t)}(\ell) = \frac{\langle \mathbf{g}_{\mathrm{cur},\ell}^{(t)}, \mathbf{g}_{\mathrm{past},\ell}^{(t)} \rangle}{\|\mathbf{g}_{\mathrm{cur},\ell}^{(t)}\| \cdot \|\mathbf{g}_{\mathrm{past},\ell}^{(t)}\|}, \tag{14}$$

$$\mathrm{conflict}^{(t)}(\ell) = \max\left(0, -\cos^{(t)}(\ell)\right). \tag{15}$$

Thus, aligned gradients incur zero conflict, while more negative cosine implies stronger conflict. To visualize whether conflict is concentrated, we normalize conflict into a per-layer share

$$\mathrm{share}^{(t)}(\ell) = \frac{\mathrm{conflict}^{(t)}(\ell)}{\sum_{\ell'=1}^{L} \mathrm{conflict}^{(t)}(\ell')}, \tag{16}$$

sort layers by $\mathrm{share}^{(t)}(\ell)$ in descending order, and plot (i) the sorted per-layer shares (bars) and (ii) their cumulative sum (curve), which yields Fig. 1(b). To summarize concentration with a scalar, we define the top-10% cumulative conflict share at boundary $t$ as

$$\mathrm{Top10Share}(t) = \sum_{k=1}^{K} s_{(k)}(t), \quad K = \lceil 0.1L \rceil, \tag{17}$$

where $L$ is the number of layers and $s_{(k)}(t)$ is the $k$-th largest conflict share at boundary $t$. A uniform distribution would give $\mathrm{Top10Share}(t) \approx 0.1$, so values substantially above $0.1$ indicate a persistent heavy-tailed pattern across boundaries. Fig. 7 shows that conflict concentration is not an artifact of a single chosen task boundary. Instead, elevated $\mathrm{Top10Share}(t)$ repeatedly appears across many boundaries, supporting that heavy-tailed layer-wise conflict is a persistent phenomenon along the continual sequence.

### A.4. Conflict-guided Reversion Ablation and Its Effect on Forgetting

We test whether the concentrated conflict layers identified in Fig. 1(b) are *causally* responsible for forgetting. At each boundary $t$ (i.e., after finishing task $t$), let $\mathcal{S}_{\mathrm{high}}(t)$ be the set of top-$K$ layers ranked by conflict share $s_\ell(t)$, where $K = \lceil rL \rceil$

and $r \in [0, 1]$ is the reverted-layer ratio. We apply *reversion* during uplink: for layers in $\mathcal{S}_{\text{high}}(t)$, we revert the transmitted adapter updates to their pre-update values (equivalently, suppressing those layers' uplink changes), while keeping the remaining layers' updates unchanged. This isolates the effect of conflict-heavy layers without discarding the full adapter structure.

We compare three settings: (i) **Full** (no reversion), (ii) **Revert-High** (revert $\mathcal{S}_{\text{high}}(t)$), and (iii) **Revert-Random** (revert $K$ uniformly random layers). Random reversion is repeated across 5 trials to compute uncertainty bands, while Revert-High is deterministic given the conflict ranking from Fig. 1(b).

To evaluate forgetting, we use backward transfer (BWT) computed from the performance matrix $A_{t,j}$, where $A_{t,j}$ is the test performance on task $j$ after finishing training task $t$. Following the standard definition, boundary-wise BWT after task $t$ is

$$\text{BWT}_t = \frac{1}{t-1} \sum_{j=1}^{t-1} (A_{t,j} - A_{j,j}), \quad t = 2, \ldots, T. \tag{18}$$

We report the *average task BWT* over boundaries

$$\text{AvgBWT} = \frac{1}{T-1} \sum_{t=2}^{T} \text{BWT}_t. \tag{19}$$

This metric captures forgetting behavior throughout the learning trajectory, rather than being dominated by a single endpoint. For reference, the *final BWT* is $\text{FinalBWT} = \text{BWT}_T$, which can be more sensitive to the last few tasks and thus less representative of typical boundary behavior. Therefore, we use $\text{AvgBWT}$ as the primary metric in Fig. 1(c).

To evaluate plasticity, we report *new-task accuracy* as the diagonal performance averaged across tasks,

$$\text{NewAcc} = \frac{1}{T} \sum_{t=1}^{T} A_{t,t}, \tag{20}$$

plotted in Fig. 1(d). Together, Fig. 1(c–d) show that reverting conflict-heavy layers improves BWT substantially with relatively small impact on new-task accuracy, while reverting random layers yields much smaller gains.

## B. Theoretical Analysis

### B.1. Proof of Conflict Attenuation

We provide the detailed derivations for Sec. 4.2, showing that attenuating the conflicting component reduces the historical-loss increment under the local quadratic proxy. Recall the replay-free local quadratic proxy for the historical loss:

$$\mathcal{L}_{<t}(\theta) \approx \text{const} + \frac{1}{2}(\theta - \bar{\theta})^{\top} \bar{F} (\theta - \bar{\theta}), \tag{21}$$

where $\bar{F} \succeq 0$ is the historical Fisher surrogate and $\bar{\theta}$ is the reference. In this paper we treat $\bar{F}$ as a diagonal matrix. Let $u := \theta - \bar{\theta}$. Consider attenuating the conflicting component of a candidate update direction by

$$v(\gamma) = v_{\text{nc}} + \gamma v_{\text{c}}, \qquad \gamma \in [0, 1], \tag{22}$$

and applying one step $\theta^{+} = \theta - \eta v(\gamma)$. Define the historical increment $\Delta(\gamma) := \mathcal{L}_{<t}(\theta^{+}) - \mathcal{L}_{<t}(\theta)$. Substituting $\theta^{+}$ into the quadratic proxy in (21) and simplifying within this proxy yields

$$\Delta(\gamma) = \frac{1}{2}\big(u - \eta v(\gamma)\big)^{\top} \bar{F} \big(u - \eta v(\gamma)\big) - \frac{1}{2} u^{\top} \bar{F} u$$
$$= -\eta\, v(\gamma)^{\top} \bar{F} u + \frac{\eta^2}{2}\, v(\gamma)^{\top} \bar{F} v(\gamma). \tag{23}$$

Let

$$S(\gamma) := v(\gamma)^{\top} \bar{F} u, \qquad a(\gamma) := v(\gamma)^{\top} \bar{F} v(\gamma), \tag{24}$$

then (23) becomes

$$\Delta(\gamma) = -\eta\, S(\gamma) + \frac{\eta^2}{2}\, a(\gamma). \tag{25}$$

Since $S(\gamma)$ is linear in $v(\gamma)$ and $a(\gamma)$ is a quadratic form, they admit the expansions

$$S(\gamma) = S_{\mathrm{nc}} + \gamma S_{\mathrm{c}}, \qquad a(\gamma) = a_{\mathrm{nc}} + 2\gamma a_{\times} + \gamma^2 a_{\mathrm{c}}, \tag{26}$$

where

$$S_{\mathrm{nc}} := v_{\mathrm{nc}}^\top \bar{F} u, \quad S_{\mathrm{c}} := v_{\mathrm{c}}^\top \bar{F} u, \quad a_{\mathrm{nc}} := v_{\mathrm{nc}}^\top \bar{F} v_{\mathrm{nc}}, \quad a_{\mathrm{c}} := v_{\mathrm{c}}^\top \bar{F} v_{\mathrm{c}} \geq 0, \quad a_{\times} := v_{\mathrm{nc}}^\top \bar{F} v_{\mathrm{c}}.$$

Here, $v_{\mathrm{c}}$ corresponds to the restriction of $v$ onto the conflicting layers, hence $S_{\mathrm{c}} = \sum_{\ell \in \mathcal{I}} S_\ell < 0$. Substituting (26) into (25) and differentiating with respect to $\gamma$ gives

$$\frac{\partial}{\partial\gamma}\Delta(\gamma) = -\eta S_{\mathrm{c}} + \eta^2\big(a_{\times} + \gamma a_{\mathrm{c}}\big). \tag{27}$$

Because $a_{\mathrm{c}} \geq 0$, the right-hand side of (27) is affine in $\gamma$ with nonnegative slope. Therefore, $\Delta(\gamma)$ is monotone nondecreasing on $\gamma \in [0, 1]$ if and only if the derivative is nonnegative at $\gamma = 0$, i.e.,

$$-\eta S_{\mathrm{c}} + \eta^2 a_{\times} \geq 0. \tag{28}$$

In particular, if $a_{\times} \geq 0$ and $S_{\mathrm{c}} < 0$, then (28) holds for any $\eta > 0$. If $a_{\times} < 0$ and $S_{\mathrm{c}} < 0$, then (28) holds whenever

$$\eta \leq \eta_\star := \frac{|S_{\mathrm{c}}|}{|a_{\times}|}. \tag{29}$$

Note that we use a diagonal Fisher surrogate and define $v_{\mathrm{c}}$ by restricting $v$ to the conflicting layers, while $v_{\mathrm{nc}}$ contains the remaining coordinates. Thus $v_{\mathrm{c}}$ and $v_{\mathrm{nc}}$ have disjoint support, and the cross-term vanishes:

$$a_{\times} = v_{\mathrm{nc}}^\top \bar{F} v_{\mathrm{c}} = \sum_{i=1}^{d} v_{\mathrm{nc},i}\, \bar{F}_i\, v_{\mathrm{c},i} = 0. \tag{30}$$

Substituting $a_{\times} = 0$ into (27) yields

$$\frac{\partial}{\partial\gamma}\Delta(\gamma) = -\eta S_{\mathrm{c}} + \eta^2 \gamma a_{\mathrm{c}}. \tag{31}$$

Since $S_{\mathrm{c}} < 0$ for any nonzero conflicting component and $a_{\mathrm{c}} \geq 0$, we have $\frac{\partial}{\partial\gamma}\Delta(\gamma) > 0$ for all $\gamma \in [0, 1]$ and any $\eta > 0$. Therefore, $\Delta(\gamma)$ is strictly increasing in $\gamma$ on $[0, 1]$, implying that decreasing $\gamma$ strictly reduces the historical-loss increment contributed by the conflicting component. $\qquad\square$

### B.2. Proof of Layer-wise Granularity

This appendix provides the detailed derivations behind the layer-wise aggregation claims used in the main text. In particular, we show that aggregating coordinate interaction scores within each LoRA layer reduces variance under weak dependence, improves an effective signal-to-noise ratio, and yields a misclassification-probability bound for the layer-wise sign decision.

Recall the coordinate interaction score (Eq. 4 in the main text)

$$s_i := v_i\, \bar{F}_i\, (\theta_i - \bar{\theta}_i),$$

and the layer-wise aggregation over layer $\ell$ (with index set $i \in \ell$ and width $n_\ell := |\ell|$) is $S_\ell := \sum_{i \in \ell} s_i$. Because $v$ and the Fisher surrogates are estimated from stochastic minibatches and heterogeneous client data, the realized scores $\{s_i\}_{i \in \ell}$ are noisy. We model this by decomposing $s_i = \mu_i + \varepsilon_i$, where $\mu_i := \mathbb{E}[s_i \mid \mathcal{H}]$ denotes the conditional mean under the current state $\mathcal{H}$, and the estimation noise satisfies $\mathbb{E}[\varepsilon_i \mid \mathcal{H}] = 0$. Define the layer-wise mean signal

$$M_\ell := \mathbb{E}[S_\ell \mid \mathcal{H}] = \sum_{i \in \ell} \mu_i = n_\ell\, \bar{\mu}_\ell, \qquad \bar{\mu}_\ell := \frac{1}{n_\ell} \sum_{i \in \ell} \mu_i. \tag{32}$$

We assume the following within-layer moment and dependence controls:

- **(H1) Zero mean noise:** $\mathbb{E}[\varepsilon_i \mid \mathcal{H}] = 0$.

- **(H2) Bounded variance:** $\mathrm{Var}(s_i \mid \mathcal{H}) = \mathrm{Var}(\varepsilon_i \mid \mathcal{H}) \leq \sigma_{\mathrm{max},\ell}^2 < \infty$ for all $i \in \ell$.

- **(H3) Bounded correlation magnitude:** for any $i \neq j$ in layer $\ell$,

$$\left|\mathrm{Corr}(s_i, s_j \mid \mathcal{H})\right| = \left|\mathrm{Corr}(\varepsilon_i, \varepsilon_j \mid \mathcal{H})\right| \leq \rho_{\mathrm{max},\ell}, \qquad \rho_{\mathrm{max},\ell} \in [0, 1).$$

These assumptions are standard weak-dependence conditions: (H2) controls per-coordinate noise, while (H3) allows correlated coordinates but prevents perfect dependence. We now bound the conditional variance of the aggregated score $S_\ell$ and the normalized score $\bar{S}_\ell := S_\ell / n_\ell$. Using the variance decomposition,

$$\mathrm{Var}(S_\ell \mid \mathcal{H}) = \sum_{i \in \ell} \mathrm{Var}(s_i \mid \mathcal{H}) + 2 \sum_{i < j} \mathrm{Cov}(s_i, s_j \mid \mathcal{H}). \tag{33}$$

To obtain a valid upper bound regardless of the sign of the covariances, we use

$$\mathrm{Var}(S_\ell \mid \mathcal{H}) \leq \sum_{i \in \ell} \mathrm{Var}(s_i \mid \mathcal{H}) + 2 \sum_{i < j} \left|\mathrm{Cov}(s_i, s_j \mid \mathcal{H})\right|. \tag{34}$$

By (H3), for any $i \neq j$,

$$\left|\mathrm{Cov}(s_i, s_j \mid \mathcal{H})\right| = \left|\mathrm{Corr}(s_i, s_j \mid \mathcal{H})\right| \sqrt{\mathrm{Var}(s_i \mid \mathcal{H}) \, \mathrm{Var}(s_j \mid \mathcal{H})} \leq \rho_{\mathrm{max},\ell} \, \sigma_{\mathrm{max},\ell}^2,$$

where we used (H2) to upper bound the variances. Substituting yields

$$\mathrm{Var}(S_\ell \mid \mathcal{H}) \leq \sum_{i \in \ell} \sigma_{\mathrm{max},\ell}^2 + 2 \sum_{i < j} \rho_{\mathrm{max},\ell} \sigma_{\mathrm{max},\ell}^2$$

$$= \sigma_{\mathrm{max},\ell}^2 \Big( n_\ell + n_\ell(n_\ell - 1)\rho_{\mathrm{max},\ell} \Big) = \sigma_{\mathrm{max},\ell}^2 \, n_\ell \Big( 1 + (n_\ell - 1)\rho_{\mathrm{max},\ell} \Big). \tag{35}$$

It is convenient to introduce an effective layer width

$$n_{\mathrm{eff},\ell} := \frac{n_\ell}{1 + (n_\ell - 1)\rho_{\mathrm{max},\ell}} \in (0, n_\ell], \tag{36}$$

so that (35) can be rewritten as

$$\mathrm{Var}(S_\ell \mid \mathcal{H}) \leq \sigma_{\mathrm{max},\ell}^2 \frac{n_\ell^2}{n_{\mathrm{eff},\ell}}. \tag{37}$$

Dividing both sides by $n_\ell^2$ yields the normalized bound claimed in the main text:

$$\mathrm{Var}\Big(\bar{S}_\ell \mid \mathcal{H}\Big) = \mathrm{Var}\Big(\tfrac{S_\ell}{n_\ell} \mid \mathcal{H}\Big) \leq \frac{\sigma_{\mathrm{max},\ell}^2}{n_{\mathrm{eff},\ell}}. \tag{38}$$

This recovers the classical $1/n_\ell$ variance reduction when $\rho_{\mathrm{max},\ell} = 0$, and degrades gracefully to the single-coordinate scale when $\rho_{\mathrm{max},\ell} \to 1$ (in which case $n_{\mathrm{eff},\ell} \to 1$). In particular, aggregation is never worse than using one coordinate, and becomes strictly better as long as correlations are not near-perfect.

Next we translate the variance reduction into a signal-to-noise improvement statement for layer-wise decisions. Define the layer-wise SNR of the summed statistic as

$$\mathrm{SNR}_\ell := \frac{|M_\ell|}{\sqrt{\mathrm{Var}(S_\ell \mid \mathcal{H})}}. \tag{39}$$

Using $M_\ell = n_\ell \bar{\mu}_\ell$ and (35),

$$\mathrm{SNR}_\ell \geq \frac{n_\ell |\bar{\mu}_\ell|}{\sigma_{\mathrm{max},\ell} \sqrt{n_\ell \big(1 + (n_\ell - 1)\rho_{\mathrm{max},\ell}\big)}} = \frac{|\bar{\mu}_\ell|}{\sigma_{\mathrm{max},\ell}} \sqrt{n_{\mathrm{eff},\ell}}. \tag{40}$$

Thus, compared with a single-coordinate scale $|\bar{\mu}_\ell|/\sigma_{\max,\ell}$, aggregation boosts SNR by at least a factor $\sqrt{n_{\mathrm{eff},\ell}}$. This is the formal sense in which layer-level scoring is more statistically stable than coordinate-wise scoring under weak dependence.

Finally, we bound the probability that the sign of the observed layer score differs from the sign of its conditional mean. This is the key event for conflict detection in our method: a layer that is truly conflicting (negative mean interaction) should not be flipped to non-conflicting by noise, and vice versa.

Assume $M_\ell \neq 0$ so that $\mathrm{sign}(M_\ell)$ is well-defined. Then

$$\{\mathrm{sign}(S_\ell) \neq \mathrm{sign}(M_\ell)\} \ \subseteq \ \{|S_\ell - M_\ell| \geq |M_\ell|\}. \tag{41}$$

Applying Chebyshev's inequality conditional on $\mathcal{H}$,

$$\mathbb{P}\big(|S_\ell - M_\ell| \geq |M_\ell| \,\big|\, \mathcal{H}\big) \ \leq \ \frac{\mathrm{Var}(S_\ell \mid \mathcal{H})}{M_\ell^2}. \tag{42}$$

Combining with (41), $M_\ell = n_\ell \bar{\mu}_\ell$, and (35) yields

$$\mathbb{P}\big(\mathrm{sign}(S_\ell) \neq \mathrm{sign}(M_\ell) \,\big|\, \mathcal{H}\big) \leq \frac{\sigma_{\max,\ell}^2 \, n_\ell \big(1 + (n_\ell - 1)\rho_{\max,\ell}\big)}{n_\ell^2 \bar{\mu}_\ell^2}$$
$$= \Big(\frac{\sigma_{\max,\ell}}{|\bar{\mu}_\ell|}\Big)^2 \frac{1}{n_{\mathrm{eff},\ell}}. \tag{43}$$

Therefore, under only finite-variance and bounded-correlation-magnitude assumptions, the probability of a wrong layer-wise sign decision decays as $1/n_{\mathrm{eff},\ell}$. $\square$

### B.3. Proof of Lemma 4.1

We prove Lemma 4.1, which states that in a normal neighborhood of $\bar{\theta}$,

$$d_F^2(\theta, \bar{\theta}) = \Delta\theta^\top \bar{F} \Delta\theta + \mathcal{E}(\theta), \qquad |\mathcal{E}(\theta)| \leq C \, \kappa_{\max} \, \|\bar{F}\| \, \|\Delta\theta\|^4,$$

where $\Delta\theta := \theta - \bar{\theta}$ and $\kappa_{\max} := \max\{\kappa, \mathcal{K}\}$ under the sectional curvature bound $-\kappa \leq \sec \leq \mathcal{K}$.

Let $(\mathcal{M}, g)$ denote the Fisher-Riemannian manifold restricted to the LoRA parameter space, and let $g(\bar{\theta}) = \bar{F} \succ 0$ denote the Fisher metric tensor evaluated at $\bar{\theta}$. Assume $\theta$ lies in a normal injectivity neighborhood of $\bar{\theta}$ so that the logarithm map is well-defined:

$$y := \log_{\bar{\theta}}(\theta) \in T_{\bar{\theta}}\mathcal{M}, \qquad \theta = \exp_{\bar{\theta}}(y).$$

Throughout, $\|\cdot\|$ denotes the Euclidean norm in a fixed coordinate chart around $\bar{\theta}$, and $\|\bar{F}\|$ is the operator norm.

In a normal neighborhood, the geodesic from $\bar{\theta}$ to $\theta$ is the radial geodesic $\gamma(t) = \exp_{\bar{\theta}}(ty)$, $t \in [0,1]$. By the defining property of the exponential map in a normal neighborhood, the geodesic distance equals the norm of the initial velocity:

$$d_F(\theta, \bar{\theta}) = \|y\|_{g(\bar{\theta})} = \sqrt{y^\top \bar{F} y}, \qquad \Rightarrow \qquad d_F^2(\theta, \bar{\theta}) = y^\top \bar{F} y. \tag{44}$$

Thus, it remains to relate the chart displacement $\Delta\theta = \theta - \bar{\theta}$ to the tangent displacement $y$. This follows from the Gauss lemma and the minimizing property of radial geodesics in a normal neighborhood; see, e.g., standard texts on Riemannian geometry (Do Carmo & Flaherty Francis, 1992).

To expand $\exp_{\bar{\theta}}(y)$ around $y = 0$, we use *normal coordinates* centered at $\bar{\theta}$. In these coordinates, $\Gamma_{jk}^i(\bar{\theta}) = 0$, hence $\ddot{\gamma}(0) = 0$ for the radial geodesic $\gamma(t) = \exp_{\bar{\theta}}(ty)$. The geodesic equation in coordinates is

$$\ddot{\gamma}^i(t) + \Gamma_{jk}^i(\gamma(t))\dot{\gamma}^j(t)\dot{\gamma}^k(t) = 0, \qquad \gamma(0) = \bar{\theta}, \ \dot{\gamma}(0) = y.$$

Taylor expanding $\gamma(t)$ at $t = 0$ and using $\ddot{\gamma}(0) = 0$ yields

$$\gamma^i(1) = \bar{\theta}^i + y^i + \frac{1}{6}\gamma^{(3),i}(0) + O(\|y\|^4), \tag{45}$$

where $\gamma^{(3)}(0)$ denotes the third derivative at $t = 0$. Differentiating the geodesic equation once and evaluating at $t = 0$ gives

$$\gamma^{(3),i}(0) = -\partial_\ell \Gamma^i_{jk}(\bar{\theta}) \, y^\ell y^j y^k.$$

Substituting into (45) and using $\theta = \gamma(1)$ yields the standard expansion of the exponential map:

$$\Delta\theta^i := \theta^i - \bar{\theta}^i = y^i - \frac{1}{6} \, \partial_\ell \Gamma^i_{jk}(\bar{\theta}) \, y^j y^k y^\ell + O(\|y\|^4). \tag{46}$$

We next express $\partial_\ell \Gamma^i_{jk}(\bar{\theta})$ via curvature. Recall the Riemann curvature tensor satisfies

$$R^i_{jkl} = \partial_k \Gamma^i_{jl} - \partial_l \Gamma^i_{jk} + \Gamma^i_{km} \Gamma^m_{jl} - \Gamma^i_{lm} \Gamma^m_{jk}.$$

At $\bar{\theta}$ in normal coordinates, $\Gamma(\bar{\theta}) = 0$, so

$$R^i_{jkl}(\bar{\theta}) = \partial_k \Gamma^i_{jl}(\bar{\theta}) - \partial_l \Gamma^i_{jk}(\bar{\theta}). \tag{47}$$

Using the symmetries of $R$ and the first Bianchi identity (Lee, 2006), one obtains the classical relation

$$\partial_\ell \Gamma^i_{jk}(\bar{\theta}) = -\frac{1}{3} \Big( R^i_{jk\ell}(\bar{\theta}) + R^i_{kj\ell}(\bar{\theta}) \Big). \tag{48}$$

Plugging (48) into (46) gives

$$\Delta\theta^i = y^i + \frac{1}{18} \Big( R^i_{jk\ell} + R^i_{kj\ell} \Big) y^j y^k y^\ell + O(\|y\|^4). \tag{49}$$

Under the sectional curvature bounds $-\kappa \leq \sec \leq \mathcal{K}$ (hence $|\sec| \leq \kappa_{\max}$ with $\kappa_{\max} := \max\{\kappa, \mathcal{K}\}$), the curvature tensor is uniformly bounded in this neighborhood. Absorbing dimension-dependent constants into $C$, we may write

$$\left\| \frac{1}{18} \Big( R^i_{jk\ell} + R^i_{kj\ell} \Big) y^j y^k y^\ell \right\| \leq C \kappa_{\max} \|y\|^3, \qquad \|O(\|y\|^4)\| \leq C\|y\|^4. \tag{50}$$

Therefore, for $\|y\|$ sufficiently small,

$$\|\Delta\theta - y\| \leq C \kappa_{\max} \|y\|^3 + C\|y\|^4 = O(\kappa_{\max}\|y\|^3). \tag{51}$$

We now make the neighborhood-shrinking step explicit. If $\kappa_{\max} = 0$, the manifold is locally flat on this neighborhood and the exponential map is linear in normal coordinates, hence $\Delta\theta = y$. Otherwise ($\kappa_{\max} > 0$), shrink the normal neighborhood so that $\|y\| \leq \min\{1, \kappa_{\max}\}$, which implies $\|y\|^4 \leq \kappa_{\max}\|y\|^3$. Then the above inequality yields $\|\Delta\theta - y\| \leq C_0 \kappa_{\max}\|y\|^3$ for a redefined constant $C_0$.

Define $e := y - \Delta\theta$. From the above bound, we have $\|e\| \leq C_0 \kappa_{\max}\|y\|^3$ for $\|y\|$ small enough. Moreover, since $\Delta\theta = y - e$, we have $\|\Delta\theta\| \geq \|y\| - \|e\|$. For sufficiently small $\|y\|$, this implies $\|y\| \leq 2\|\Delta\theta\|$ and hence

$$\|e\| \leq C \kappa_{\max} \|y\|^3 \leq C \kappa_{\max} (2\|\Delta\theta\|)^3 \leq C' \kappa_{\max} \|\Delta\theta\|^3. \tag{52}$$

Thus,

$$y = \Delta\theta + e \qquad \text{with} \qquad \|e\| \leq C' \kappa_{\max} \|\Delta\theta\|^3. \tag{53}$$

Using (44) and (53),

$$\begin{aligned} d_F^2(\theta, \bar{\theta}) = y^\top \bar{F} y &= (\Delta\theta + e)^\top \bar{F} (\Delta\theta + e) \\ &= \Delta\theta^\top \bar{F} \Delta\theta + 2e^\top \bar{F} \Delta\theta + e^\top \bar{F} e. \end{aligned} \tag{54}$$

Define the remainder

$$\mathcal{E}(\theta) := 2e^\top \bar{F} \Delta\theta + e^\top \bar{F} e.$$

By Cauchy–Schwarz and the operator norm bound,

$$|2e^\top \bar{F} \Delta\theta| \leq 2\|\bar{F}\| \, \|e\| \, \|\Delta\theta\| \leq 2\|\bar{F}\| \cdot (C'\kappa_{\max}\|\Delta\theta\|^3) \cdot \|\Delta\theta\| \leq C_1 \kappa_{\max} \|\bar{F}\| \|\Delta\theta\|^4,$$

and similarly

$$|e^\top \bar{F} e| \leq \|\bar{F}\| \, \|e\|^2 \leq \|\bar{F}\| \cdot (C'\kappa_{\max})^2 \|\Delta\theta\|^6 \leq C_2 \kappa_{\max} \|\bar{F}\| \|\Delta\theta\|^4 \quad \text{(for } \|\Delta\theta\| \text{ small).}$$

Combining the two bounds yields

$$|\mathcal{E}(\theta)| \leq C \kappa_{\max} \|\bar{F}\| \|\Delta\theta\|^4, \tag{55}$$

for a (redefined) universal constant $C$. Substituting (55) into (54) completes the proof. $\square$

## B.4. Proof of Theorem 4.2

We prove Theorem 4.2 using the same notation as Appendix B.3. Let $R_\epsilon := \sqrt{2\epsilon}$ and, when $\mathcal{K} > 0$, assume $R_\epsilon < \pi/(2\sqrt{\mathcal{K}})$. Define the intrinsic distance

$$\rho(\theta) := d_F(\theta, \bar{\theta}), \qquad \Delta\theta := \theta - \bar{\theta},$$

and the quadratic proxy used throughout the main text

$$r^2(\theta) := \Delta\theta^\top \bar{F} \Delta\theta, \qquad \bar{F} := g(\bar{\theta}).$$

Define the (local) basin

$$\mathcal{B}_\epsilon(\bar{\theta}) := \{\theta \in \mathcal{N} : \rho(\theta)^2 \le 2\epsilon\}.$$

We work throughout within the normal neighborhood $\mathcal{N}$ under the bounded sectional-curvature assumption of Theorem 4.2, so all comparison factors in the sequel are understood in this local regime. Throughout this proof we assume $\bar{F} \succeq \varepsilon_F I$ for some $\varepsilon_F > 0$, hence $\lambda_{\min}(\bar{F}) \ge \varepsilon_F > 0$.

By Lemma 4.1, for all $\theta$ sufficiently close to $\bar{\theta}$,

$$\left| \rho(\theta)^2 - r^2(\theta) \right| \le C_\kappa \|\Delta\theta\|^4, \qquad C_\kappa = C\,\kappa_{\max}\,\|\bar{F}\|. \tag{56}$$

Moreover, since $r^2(\theta) \ge \lambda_{\min}\|\Delta\theta\|^2$, we have $\|\Delta\theta\|^4 \le \lambda_{\min}^{-2} r^4(\theta)$ and thus

$$\left| \rho(\theta)^2 - r^2(\theta) \right| \le \widetilde{C}_\kappa\, r^4(\theta), \qquad \widetilde{C}_\kappa := C_\kappa \lambda_{\min}^{-2}. \tag{57}$$

Fix $R \in (0, R_0]$ such that the geodesic ball $B_R^{\mathrm{geo}}(\bar{\theta}) := \{\theta : \rho(\theta) \le R\}$ is contained in a normal neighborhood where (56)–(57) hold. For $\theta \in B_R^{\mathrm{geo}}(\bar{\theta})$, combining $\rho(\theta)^2 \le R^2$ with (56) gives

$$R^2 \ge \rho(\theta)^2 \ge r^2(\theta) - C_\kappa\|\Delta\theta\|^4 \ge \lambda_{\min}\|\Delta\theta\|^2 - C_\kappa\|\Delta\theta\|^4.$$

Choose $R_0$ small enough so that $C_\kappa(c_0 R_0)^2 \le \frac{1}{2}\lambda_{\min}$ for a numerical constant $c_0$, and absorb the quartic term to obtain

$$\|\Delta\theta\| \le c\,R, \qquad \forall\,\theta \in B_R^{\mathrm{geo}}(\bar{\theta}), \tag{58}$$

for some constant $c > 0$ (depending only on $\lambda_{\min}$ and $C_\kappa$). On $B_R^{\mathrm{geo}}(\bar{\theta})$, (58) implies $\|\Delta\theta\|^4 \le (c^2 R^2)\|\Delta\theta\|^2$. Using $r^2(\theta) \ge \lambda_{\min}\|\Delta\theta\|^2$, we obtain

$$\|\Delta\theta\|^4 \le (c^2 R^2)\lambda_{\min}^{-1}\, r^2(\theta), \qquad \theta \in B_R^{\mathrm{geo}}(\bar{\theta}). \tag{59}$$

Plugging (59) into (56) yields

$$\left| \rho(\theta)^2 - r^2(\theta) \right| \le \delta_R\, r^2(\theta), \qquad \theta \in B_R^{\mathrm{geo}}(\bar{\theta}), \tag{60}$$

where

$$\delta_R := C_{\mathrm{rel}}\,\kappa_{\max}R^2, \qquad C_{\mathrm{rel}} := C\,\|\bar{F}\|\,c^2\,\lambda_{\min}^{-1}.$$

In particular, for $R \le R_0$ we may assume $\delta_R < 1$. Define the proxy ellipsoid $E_R := \{\theta : r^2(\theta) \le R^2\}$. From (60) we obtain the explicit two-sided inclusions:

$$B_R^{\mathrm{geo}}(\bar{\theta}) \subset E_{R/\sqrt{1-\delta_R}}, \qquad E_{R/\sqrt{1+\delta_R}} \subset B_R^{\mathrm{geo}}(\bar{\theta}). \tag{61}$$

Equivalently, within radius $R$, the intrinsic ball and the proxy ellipsoid differ only by the multiplicative radius distortion $\sqrt{(1 \pm \delta_R)^{-1}} = 1 + O(\kappa_{\max}R^2)$.

Let $\mathrm{KL}(\theta) := \mathrm{KL}(p_\theta\|p_{\bar{\theta}})$. Under standard regularity assumptions, a Taylor expansion at $\bar{\theta}$ yields

$$2\,\mathrm{KL}(p_\theta\|p_{\bar{\theta}}) = r^2(\theta) + \frac{1}{3}\,T_{ijk}(\bar{\theta})\,\Delta\theta^i\Delta\theta^j\Delta\theta^k + O(\|\Delta\theta\|^4), \tag{62}$$

where $T_{ijk}(\bar{\theta})$ is the totally symmetric Amari–Chentsov tensor (Amari & Nagaoka, 2000). Together with (56), this shows that both $2\,\mathrm{KL}(p_\theta\|p_{\bar{\theta}})$ and $\rho(\theta)^2 = d_F^2(\theta, \bar{\theta})$ share the same quadratic term $r^2(\theta)$. We do not use KL in the remainder.

We apply the classical Hessian comparison theorem (Jost, 2005) to the distance function $\rho(\theta) = d_F(\theta, \bar{\theta})$ on $\mathcal{B}_\epsilon(\bar{\theta})$, under sectional curvature bounds $-\kappa \leq \sec \leq \mathcal{K}$. For $c \in \mathbb{R}$ and $s > 0$ define

$$\mathrm{sn}_c(s) := \begin{cases} \frac{1}{\sqrt{c}} \sin(\sqrt{c}\, s), & c > 0, \\ s, & c = 0, \\ \frac{1}{\sqrt{-c}} \sinh(\sqrt{-c}\, s), & c < 0, \end{cases} \quad \mathrm{cs}_c(s) := \frac{d}{ds}\, \mathrm{sn}_c(s), \quad \mathrm{ct}_c(s) := \frac{\mathrm{cs}_c(s)}{\mathrm{sn}_c(s)}.$$

Define the curvature comparison factors

$$d_{+\mathcal{K}}(s) := s\, \mathrm{ct}_{+\mathcal{K}}(s), \qquad d_{-\kappa}(s) := s\, \mathrm{ct}_{-\kappa}(s). \tag{63}$$

The Hessian comparison theorem gives, for $\theta \neq \bar{\theta}$,

$$\mathrm{ct}_{+\mathcal{K}}(\rho)\, (g - d\rho \otimes d\rho) \ \preceq\ \mathrm{Hess}\, \rho \ \preceq\ \mathrm{ct}_{-\kappa}(\rho)\, (g - d\rho \otimes d\rho). \tag{64}$$

Let $\phi(\theta) := \frac{1}{2}\rho(\theta)^2 = \frac{1}{2}d_F^2(\theta, \bar{\theta})$. Using $\mathrm{Hess}\, \phi = d\rho \otimes d\rho + \rho\, \mathrm{Hess}\, \rho$ and (64), we obtain

$$\left(1 - d_{+\mathcal{K}}(\rho)\right) d\rho \otimes d\rho + d_{+\mathcal{K}}(\rho)\, g \ \preceq\ \mathrm{Hess}\left(\tfrac{1}{2}\rho^2\right) \ \preceq\ \left(1 - d_{-\kappa}(\rho)\right) d\rho \otimes d\rho + d_{-\kappa}(\rho)\, g. \tag{65}$$

For $\mathcal{K} > 0$ and $\rho < \pi/(2\sqrt{\mathcal{K}})$ we have $0 < d_{+\mathcal{K}}(\rho) \leq 1$, hence $1 - d_{+\mathcal{K}}(\rho) \geq 0$. For $\kappa > 0$ we have $d_{-\kappa}(\rho) \geq 1$, hence $1 - d_{-\kappa}(\rho) \leq 0$. Since $d\rho \otimes d\rho \succeq 0$, dropping the rank-one terms in (65) yields

$$d_{+\mathcal{K}}(\rho(\theta))\, g_\theta \ \preceq\ \mathrm{Hess}\left(\tfrac{1}{2}\rho^2\right)(\theta) \ \preceq\ d_{-\kappa}(\rho(\theta))\, g_\theta, \qquad \theta \neq \bar{\theta}. \tag{66}$$

On $[0, \pi/(2\sqrt{\mathcal{K}}))$, $d_{+\mathcal{K}}$ is non-increasing; on $[0, \infty)$, $d_{-\kappa}$ is non-decreasing. For $\theta \in \mathcal{B}_\epsilon(\bar{\theta})$, $\rho(\theta) \leq R_\epsilon$, hence

$$d_{+\mathcal{K}}(\rho(\theta)) \geq d_{+\mathcal{K}}(R_\epsilon), \qquad d_{-\kappa}(\rho(\theta)) \leq d_{-\kappa}(R_\epsilon).$$

Applying these bounds to (66) yields the uniform sandwich

$$d_{+\mathcal{K}}(R_\epsilon)\, g_\theta \ \preceq\ \mathrm{Hess}\left(\tfrac{1}{2}\rho^2\right)(\theta) \ \preceq\ d_{-\kappa}(R_\epsilon)\, g_\theta, \qquad \forall \theta \in \mathcal{B}_\epsilon(\bar{\theta}) \setminus \{\bar{\theta}\}. \tag{67}$$

We will use the factors $d_{+\mathcal{K}}(R_\epsilon)$ and $d_{-\kappa}(R_\epsilon)$ to parameterize the curvature dependence in the comparison constants of Theorem 4.2.

Choose $\epsilon > 0$ small enough so that $R_\epsilon \leq R_0$ and $\delta_{R_\epsilon} < 1$, and define $\delta_\epsilon := \delta_{R_\epsilon} = C_{\mathrm{rel}}\kappa_{\max}R_\epsilon^2$. Then for every $\theta \in \mathcal{B}_\epsilon(\bar{\theta}) = B_{R_\epsilon}^{\mathrm{geo}}(\bar{\theta})$,

$$(1 - \delta_\epsilon)\, r^2(\theta) \ \leq\ \rho(\theta)^2 \ \leq\ (1 + \delta_\epsilon)\, r^2(\theta). \tag{68}$$

On the admissible range of $R_\epsilon$ (in particular, $R_\epsilon < \pi/(2\sqrt{\mathcal{K}})$ when $\mathcal{K} > 0$), the comparison factors satisfy

$$0 < d_{+\mathcal{K}}(R_\epsilon) \leq 1, \qquad d_{-\kappa}(R_\epsilon) \geq 1.$$

Define

$$\psi_{+\mathcal{K}}(R_\epsilon) := d_{+\mathcal{K}}(R_\epsilon)\,(1 - \delta_\epsilon), \qquad \psi_{-\kappa}(R_\epsilon) := d_{-\kappa}(R_\epsilon)\,(1 + \delta_\epsilon). \tag{69}$$

Then (68) immediately implies

$$\psi_{+\mathcal{K}}(R_\epsilon)\, r^2(\theta) = d_{+\mathcal{K}}(R_\epsilon)(1 - \delta_\epsilon)\, r^2(\theta) \leq (1 - \delta_\epsilon)\, r^2(\theta) \leq \rho(\theta)^2,$$

and similarly,

$$\rho(\theta)^2 \leq (1 + \delta_\epsilon)\, r^2(\theta) \leq d_{-\kappa}(R_\epsilon)(1 + \delta_\epsilon)\, r^2(\theta) = \psi_{-\kappa}(R_\epsilon)\, r^2(\theta).$$

Therefore, for all $\theta \in \mathcal{B}_\epsilon(\bar{\theta})$ we have

$$\psi_{+\mathcal{K}}(R_\epsilon)\, r^2(\theta) \ \leq\ d_F^2(\theta, \bar{\theta}) \ \leq\ \psi_{-\kappa}(R_\epsilon)\, r^2(\theta),$$

which is exactly Theorem 4.2. Using the Taylor expansions around 0,

$$d_{+\mathcal{K}}(s) = 1 - \frac{\mathcal{K}}{3}s^2 + O(\mathcal{K}^2 s^4), \qquad d_{-\kappa}(s) = 1 + \frac{\kappa}{3}s^2 + O(\kappa^2 s^4),$$

and $\delta_\epsilon = O(\kappa_{\max} R_\epsilon^2)$, we obtain

$$\psi_{+\mathcal{K}}(R_\epsilon) = 1 - \frac{\mathcal{K}}{3}R_\epsilon^2 \ + \ O(\kappa_{\max} R_\epsilon^2) \ + \ O(\mathcal{K}^2 R_\epsilon^4),$$

$$\psi_{-\kappa}(R_\epsilon) = 1 + \frac{\kappa}{3}R_\epsilon^2 \ + \ O(\kappa_{\max} R_\epsilon^2) \ + \ O(\kappa^2 R_\epsilon^4).$$

This completes the proof. $\square$

### B.5. Proof of Proposition 4.3

We prove Proposition 4.3. Let $R_\epsilon := \sqrt{2\epsilon}$ and $\beta_\epsilon := \psi_{-\kappa}(R_\epsilon) \geq 1$ be the upper comparison factor from Theorem 4.2. Define the geodesic basin and the certified set

$$\mathcal{B}_\epsilon(\bar{\theta}) := \{\theta \in \mathcal{N} : \ d_F(\theta, \bar{\theta}) \leq R_\epsilon\}, \qquad \mathcal{C}_\epsilon := \{\theta \in \mathcal{N} : \ \beta_\epsilon \, r^2(\theta) \leq R_\epsilon^2\}.$$

Assume $\mathcal{C}_\epsilon \subseteq \mathcal{B}_\epsilon(\bar{\theta})$, so that Theorem 4.2 applies on $\mathcal{C}_\epsilon$. Let $\theta^+$ be an updated iterate. If $\theta^+ \in \mathcal{C}_\epsilon$, then by Theorem 4.2,

$$d_F^2(\theta^+, \bar{\theta}) \ \leq \ \beta_\epsilon \, r^2(\theta^+) \ \leq \ R_\epsilon^2,$$

hence $\theta^+ \in \mathcal{B}_\epsilon(\bar{\theta})$. Therefore it suffices to enforce the computable certificate

$$\beta_\epsilon \, r^2(\theta^+) \ \leq \ R_\epsilon^2. \tag{70}$$

Consider a layer-restricted update $\theta^+ = \theta - \eta v$ where only layer $\ell$ is modified (i.e., $v_j = 0$ for $j \neq \ell$). This restriction yields a closed-form certified step size; in the full algorithm we apply the same controller block-wise (and combine it with the budgeted selection step), using the additivity of the diagonal Fisher quadratic proxy across layers.

Using the block-diagonal proxy $r^2(\theta) = \sum_{j=1}^{L}(\theta_j - \bar{\theta}_j)^\top \bar{F}_j(\theta_j - \bar{\theta}_j)$, we obtain the exact expansion

$$r^2(\theta^+) = r^2(\theta) - 2\eta S_\ell + \eta^2 Q_\ell, \tag{71}$$

where

$$S_\ell := v_\ell^\top \bar{F}_\ell(\theta_\ell - \bar{\theta}_\ell), \qquad Q_\ell := v_\ell^\top \bar{F}_\ell v_\ell \ \geq \ 0.$$

We allow two nonnegative margins $\vartheta \geq 0$ and $\varsigma \geq 0$ and use the conservative replacements

$$r^2(\theta) \ \leq \ r^2(\theta) + \varsigma, \qquad S_\ell \ \geq \ S_\ell - \vartheta.$$

Substituting into (71) yields the certified upper bound

$$r^2(\theta^+) \ \leq \ r^2(\theta) + \varsigma \ - \ 2\eta(S_\ell - \vartheta) \ + \ \eta^2 Q_\ell. \tag{72}$$

Plugging (72) into the sufficient condition (70), define

$$\delta_\epsilon \ := \ R_\epsilon^2 - \beta_\epsilon\big(r^2(\theta) + \varsigma\big).$$

Then (70) is implied by the quadratic inequality

$$\beta_\epsilon Q_\ell \, \eta^2 \ - \ 2\beta_\epsilon(S_\ell - \vartheta)\,\eta \ - \ \delta_\epsilon \ \leq \ 0. \tag{73}$$

If $\delta_\epsilon < 0$, no nonnegative step can satisfy (70), so we set $\eta_{\text{safe}} := 0$. If $Q_\ell = 0$, the closed-form expression below is undefined; we conservatively set $\eta_{\text{safe}} := 0$ (degenerate/near-null Fisher direction). Assume henceforth that $\delta_\epsilon \geq 0$ and $Q_\ell > 0$.

The roots of (73) are

$$\eta_\pm = \frac{(S_\ell - \vartheta) \pm \sqrt{(S_\ell - \vartheta)^2 + \frac{Q_\ell}{\beta_\epsilon}\delta_\epsilon}}{Q_\ell}.$$

In a conflicting layer we have $S_\ell - \vartheta < 0$, hence $\eta_- < 0$ and the set of feasible nonnegative steps is $[0, \eta_+]$. Therefore the maximal certified step size is

$$\eta_{\text{safe}} = \eta_+ = \frac{1}{Q_\ell}\Big((S_\ell - \vartheta) + \sqrt{(S_\ell - \vartheta)^2 + \frac{Q_\ell}{\beta_\epsilon}\delta_\epsilon}\Big). \tag{74}$$

For any $\eta \in [0, \eta_{\text{safe}}]$, inequality (73) holds, hence $\beta_\epsilon r^2(\theta^+) \leq R_\epsilon^2$, i.e., $\theta^+ \in \mathcal{C}_\epsilon$. Under $\mathcal{C}_\epsilon \subseteq \mathcal{B}_\epsilon(\bar\theta)$, we conclude $\theta^+ \in \mathcal{B}_\epsilon(\bar\theta)$. This completes the proof. $\square$

### B.6. A Strict Plug-in Upper Bound for the Inflation Factor $\beta_\epsilon$

We work at a fixed layer $\ell$. Let $\bar\theta$ be the reference parameters for this layer and $\bar F$ be the diagonal Fisher surrogate at $\bar\theta$. Define the quadratic proxy

$$r^2(\theta) := \|\theta - \bar\theta\|_{\bar F}^2 = (\theta - \bar\theta)^\top \bar F(\theta - \bar\theta).$$

Let $R_\epsilon := \sqrt{2\epsilon}$ and define the intrinsic basin

$$B_\epsilon(\bar\theta) := \{\theta \in \mathcal{N} : \ d_F(\theta, \bar\theta) \leq R_\epsilon\}.$$

By Theorem 4.2, on $B_\epsilon(\bar\theta)$ we have the upper comparison bound

$$d_F^2(\theta, \bar\theta) \leq \beta_\epsilon\, r^2(\theta), \qquad \beta_\epsilon \geq 1. \tag{75}$$

Theorem 4.2 provides one valid choice $\beta_\epsilon = \psi_{-\kappa}(R_\epsilon)$; in what follows we construct a computable layer-wise upper bound under the diagonal Fisher approximation. We enforce (75) via the certificate set

$$\mathcal{C}_\epsilon := \{\theta \in \mathcal{N} : \ \beta_\epsilon r^2(\theta) \leq R_\epsilon^2\}.$$

To avoid circularity, we assume (as in other local comparison arguments) that $\mathcal{C}_\epsilon \subseteq B_\epsilon(\bar\theta)$, so that (75) is valid whenever the certificate holds. The assumption $\mathcal{C}_\epsilon \subseteq B_\epsilon(\bar\theta)$ simply enforces a local regime where (75) applies. By Lemma B.1, using any conservative proxy $\hat\beta \geq \beta_\epsilon$ can only shrink the certified step, which motivates (77).

Consider the quadratic certificate used in Proposition 4.3:

$$\beta\Big(r^2(\theta) - 2\eta(S_\ell - \vartheta) + \eta^2 Q_\ell + \varsigma\Big) \ \leq \ R_\epsilon^2, \tag{76}$$

where $S_\ell, Q_\ell$ are defined in Proposition 4.3 and $\vartheta, \varsigma \geq 0$ are margins. Let $\eta_{\text{safe}}(\beta)$ be the maximal $\eta \geq 0$ satisfying (76).

**Lemma B.1.** $\eta_{\text{safe}}(\beta)$ is non-increasing in $\beta$.

*Proof.* Let $\theta^+ := \theta - \eta v$. Using $r^2(\theta^+) = r^2(\theta) - 2\eta S_\ell + \eta^2 Q_\ell$, the bracket in (76) equals $r^2(\theta^+) + 2\eta\vartheta + \varsigma \geq 0$. Hence for fixed $\eta$, the LHS of (76) is non-decreasing in $\beta$, implying the feasible set of $\eta$ shrinks as $\beta$ increases. Therefore the maximal feasible $\eta_{\text{safe}}(\beta)$ is non-increasing. $\square$

Let $\hat\beta \geq \beta_\epsilon$ and define $\eta_{\text{safe}}(\hat\beta)$ as the maximal $\eta \geq 0$ satisfying (76) with $\beta = \hat\beta$ (equivalently, the closed form in Proposition 4.3 with $\beta_\epsilon$ replaced by $\hat\beta$ and the residual recomputed accordingly).

**Proposition B.2.** *Let $\eta := \eta_{\text{safe}}(\hat\beta)$ and $\theta^+ := \theta - \eta v$. Then: (i) $\eta \leq \eta_{\text{safe}}(\beta_\epsilon)$; and (ii) $\beta_\epsilon r^2(\theta^+) \leq R_\epsilon^2$, hence $\theta^+ \in \mathcal{C}_\epsilon$. Under $\mathcal{C}_\epsilon \subseteq B_\epsilon(\bar\theta)$, this implies $\theta^+ \in B_\epsilon(\bar\theta)$ and $d_F^2(\theta^+, \bar\theta) \leq R_\epsilon^2(= 2\epsilon)$.*

*Proof.* Part (i) follows from Lemma B.1 since $\hat\beta \geq \beta_\epsilon$. For (ii), since $\eta$ satisfies (76) with $\beta = \hat\beta$ and the bracket is nonnegative,

$$\hat\beta(\cdots) \leq R_\epsilon^2 \quad \Rightarrow \quad \beta_\epsilon(\cdots) \leq R_\epsilon^2 \quad \Rightarrow \quad \beta_\epsilon r^2(\theta^+) \leq R_\epsilon^2.$$

The final claim follows by $\mathcal{C}_\epsilon \subseteq B_\epsilon(\bar\theta)$ and (75). $\square$

The factor $\beta_\epsilon$ is not directly observable without access to curvature bounds. Instead, we upper-bound the local inflation of the Fisher metric relative to the reference metric $\bar{F}$ on the considered neighborhood. Concretely, let $F_{\mathrm{curr}}$ denote any diagonal matrix that satisfies (entrywise on layer $\ell$) $F_{\mathrm{diag}}(\theta) \preceq F_{\mathrm{curr}}$ for all $\theta \in \mathcal{C}_\epsilon$. Under the diagonal Fisher approximation we write $F(\theta) \equiv F_{\mathrm{diag}}(\theta)$, so that for any such $\theta$ and any $\Delta\theta$ we have $\Delta\theta^\top F(\theta)\Delta\theta \le \Delta\theta^\top F_{\mathrm{curr}}\Delta\theta$. Under diagonal PSD matrices, the associated layer-wise generalized-eigenvalue ratio admits a Rayleigh-quotient characterization and reduces to a coordinate-wise maximum. Defining

$$\hat{\beta}_\ell := \max_{i \in \ell} \frac{\max\{\bar{F}_i, F_{\mathrm{curr},i}\}}{\bar{F}_i} \qquad (\bar{F}_i > 0), \tag{77}$$

we have $\Delta\theta^\top F_{\mathrm{curr}}\Delta\theta \le \hat{\beta}_\ell \, \Delta\theta^\top \bar{F} \Delta\theta$. Combining the two inequalities yields the pointwise domination $F(\theta) \preceq \hat{\beta}_\ell \bar{F}$ on $\mathcal{C}_\epsilon$. Finally, for any $\theta \in \mathcal{C}_\epsilon$ and any absolutely continuous curve $\gamma : [0,1] \to \mathcal{C}_\epsilon$ with $\gamma(0) = \bar{\theta}$ and $\gamma(1) = \theta$, the induced curve length satisfies

$$L_F(\gamma) = \int_0^1 \sqrt{\dot{\gamma}(t)^\top F(\gamma(t))\dot{\gamma}(t)} \, dt \le \sqrt{\hat{\beta}_\ell} \int_0^1 \sqrt{\dot{\gamma}(t)^\top \bar{F} \dot{\gamma}(t)} \, dt = \sqrt{\hat{\beta}_\ell} \, L_{\bar{F}}(\gamma).$$

Taking the infimum over all such curves gives $d_F(\theta, \bar{\theta}) \le \sqrt{\hat{\beta}_\ell} \, d_{\bar{F}}(\theta, \bar{\theta}) = \sqrt{\hat{\beta}_\ell} \, \|\theta - \bar{\theta}\|_{\bar{F}}$, hence $d_F^2(\theta, \bar{\theta}) \le \hat{\beta}_\ell \, r^2(\theta)$ on $\mathcal{C}_\epsilon$. Therefore $\hat{\beta}_\ell$ is a valid choice of $\beta_\epsilon$ in (75). We use the one-sided inflation bound $\max\{\bar{F}_i, F_{\mathrm{curr},i}\}$ to ensure $\hat{\beta}_\ell \ge 1$. $\quad\square$

## B.7. Derivations for the benefit and forgetting proxies and telescoping bounds

In particular, $r^2(\theta)$ denotes the maintained historical quadratic surrogate, and $\beta_\varepsilon r^2(\theta) \le R_\varepsilon^2$ is the Safe-Basin certificate, with $R_\varepsilon := \sqrt{2\varepsilon}$. We also write $r_{\ell,e}^2$ for the layer-wise value of $r^2(\cdot)$ at local step $e$ of layer $\ell$.

Consider a task loss $f(\theta)$ and a layer-wise update $\theta_{e+1}^{(\ell)} = \theta_e^{(\ell)} - \eta_{\ell,e} v_e^{(\ell)}$, where $g_{\ell,e} := \nabla_{\theta^{(\ell)}} f(\theta_e)$. A second-order (directional) surrogate of the one-step improvement along $v_e^{(\ell)}$ takes the form

$$f(\theta_e) - f(\theta_{e+1}) \approx \eta_{\ell,e} \, g_{\ell,e}^\top v_e^{(\ell)} - \frac{1}{2}\eta_{\ell,e}^2 \, \mathcal{C}_{\ell,e}, \tag{78}$$

where the second-order term can be written as a directional curvature

$$\mathcal{C}_{\ell,e} := \left(v_e^{(\ell)}\right)^\top H_{\ell,e} \, v_e^{(\ell)} \ge 0,$$

with $H_{\ell,e}$ the (local) Hessian of $f$ restricted to layer $\ell$ (or any PSD curvature surrogate). Evaluating $\mathcal{C}_{\ell,e}$ is expensive, so we replace it by a cheap scalar proxy that scales with the directional first-order signal and preserves nonnegativity:

$$\mathcal{C}_{\ell,e} \approx \widehat{\mathcal{C}}_{\ell,e} := g_{\ell,e}^\top v_e^{(\ell)}, \qquad \widehat{\mathcal{C}}_{\ell,e} \ge 0 \text{ whenever } g_{\ell,e}^\top v_e^{(\ell)} \ge 0.$$

Substituting $\widehat{\mathcal{C}}_{\ell,e}$ into the second-order surrogate yields the practical benefit score

$$B_{\ell,e} := \left[\eta_{\ell,e} \, g_{\ell,e}^\top v_e^{(\ell)} - \frac{1}{2}\eta_{\ell,e}^2 \, g_{\ell,e}^\top v_e^{(\ell)}\right]_+ = \left[\left(\eta_{\ell,e} - \tfrac{1}{2}\eta_{\ell,e}^2\right) g_{\ell,e}^\top v_e^{(\ell)}\right]_+. \tag{79}$$

Fix a layer $\ell$ and consider $\theta_{e+1}^{(\ell)} = \theta_e^{(\ell)} - \eta_{\ell,e} v_e^{(\ell)}$. Let $S_{\ell,e}$ and $Q_{\ell,e}$ be the same two layer-wise scalars as in the main text (evaluated at step $e$), so that the quadratic expansion of the maintained surrogate gives

$$r_{\ell,e+1}^2 = r_{\ell,e}^2 - 2\eta_{\ell,e} S_{\ell,e} + \eta_{\ell,e}^2 Q_{\ell,e}. \tag{80}$$

Define the (un-clipped) per-step increment

$$\Delta\Phi_{\ell,e} := -\eta_{\ell,e} S_{\ell,e} + \tfrac{1}{2}\eta_{\ell,e}^2 Q_{\ell,e} = \tfrac{1}{2}\left(r_{\ell,e+1}^2 - r_{\ell,e}^2\right). \tag{81}$$

Summing over $e = 0, \dots, E-1$ yields the exact telescoping identity

$$\sum_{e=0}^{E-1} \Delta\Phi_{\ell,e} = \tfrac{1}{2}\left(r_{\ell,E}^2 - r_{\ell,0}^2\right).$$

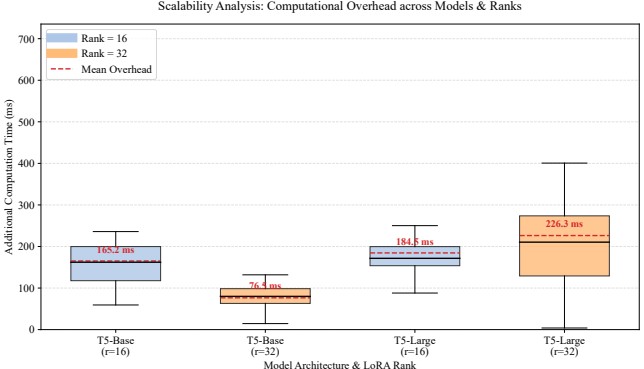

*Figure 8.* **Scalability Analysis of Computational Overhead.** Box plots aggregating runtime data from 4 distinct clients for varying model sizes (T5-Base vs. T5-Large) and LoRA ranks ($r = 16, 32$). The red dashed lines indicate the mean additional time per batch.

The round-level forgetting proxy clips *after* summation:

$$\Phi_{\ell,m} := \Big[ \sum_{e=0}^{E-1} \Delta\Phi_{\ell,e} \Big]_+ = \tfrac{1}{2}\big(r_{\ell,E}^2 - r_{\ell,0}^2\big)_+, \tag{82}$$

which is the endpoint form used in the paper.

For clarity we present a global version; the layer-wise statement is identical. Let $\theta_{e+1} = \theta_e - \eta_e v_e$ and denote $r_e^2 := r^2(\theta_e)$. Define

$$\Delta\Phi_e := -\eta_e v_e^\top \bar{F}(\theta_e - \bar{\theta}) + \tfrac{1}{2}\eta_e^2 v_e^\top \bar{F} v_e = \tfrac{1}{2}(r_{e+1}^2 - r_e^2),$$

so that

$$\sum_{e=0}^{E-1} \Delta\Phi_e = \tfrac{1}{2}(r_E^2 - r_0^2), \qquad \Phi^{\text{total}} := \Big[ \sum_{e=0}^{E-1} \Delta\Phi_e \Big]_+ = \tfrac{1}{2}(r_E^2 - r_0^2)_+. \tag{83}$$

Assume Safe-Basin maintains $\beta_\varepsilon r_e^2 \le R_\varepsilon^2$ for all $e$. Then $r_e^2 \le R_\varepsilon^2/\beta_\varepsilon$ and hence

$$\Phi^{\text{total}} \le \tfrac{1}{2}\Big(\frac{R_\varepsilon^2}{\beta_\varepsilon} - r_0^2\Big)_+ \le \tfrac{1}{2}\frac{R_\varepsilon^2}{\beta_\varepsilon} = \frac{\varepsilon}{\beta_\varepsilon} \le \varepsilon,$$

since $\beta_\varepsilon \ge 1$.

## C. Empirical Analysis of Computational Overhead

In the main text, we stated that *RieSelect* introduces minimal computational overhead with $O(d)$ complexity. This appendix provides supporting empirical evidence for this claim by analyzing the computational cost from two complementary perspectives: scalability across model architectures (Macro-view) and stability across training iterations (Micro-view). The overhead is defined as the time difference $\Delta t = t_{\text{RieSelect}} - t_{\text{Baseline}}$ per training batch, where $t_{\text{Baseline}}$ denotes the time consumed by a single LoRA training step with the same batch size.

### C.1. Scalability Analysis (Macro-view)

We first investigate how the computational overhead scales with the model size and LoRA rank. Fig. 8 aggregates runtime data from multiple heterogeneous clients across two distinct model architectures: T5-Base and T5-Large, with LoRA ranks set to $r = 16$ and $r = 32$.

Despite the theoretical linear dependence on parameter count $d$, the empirical results reveal a favorable "flat" cost curve. As illustrated in Fig. 8, while the model size increases by approximately $3.5\times$ (from Base to Large), the absolute computational overhead remains stable within the millisecond-level range (e.g., increasing marginally from 165.2 ms to 184.5 ms for $r = 16$). Crucially, the relative overhead decreases as model complexity grows: for the largest evaluated model (T5-Large, $r = 32$), the overhead constitutes only 16.15% of the baseline training time (226.3 ms vs. 1400.9 ms), compared to $\approx 32.4\%$

for T5-Base ($r = 16$). This trend indicates that our element-wise geometric operations scale much more efficiently than the backbone's heavy gradient backpropagation. Consequently, the total latency for large models is increasingly dominated by the backbone training itself rather than our selection mechanism. This property confirms the high scalability of our method and suggests it becomes even more computationally economical when applied to large-scale foundation models.

### C.2. Runtime Stability Analysis (Micro-view)

Beyond average costs, reliability in federated edge training requires computational stability to prevent unexpected latency spikes. Fig. 9 visualizes the step-wise overhead for a representative client training the largest evaluated model (T5-Large, $r = 32$). The scatter plot demonstrates that the overhead is computationally deterministic. The data points cluster tightly around the mean ($\approx 159$ ms), with the majority falling within the narrow stability region ($\pm 1\sigma$). The absence of significant outliers or upward drifts over time confirms that our conflict detection and step-size adjustment mechanisms do not introduce data-dependent jitter. This ensures a predictable training timeline which is essential for synchronous federated learning settings.

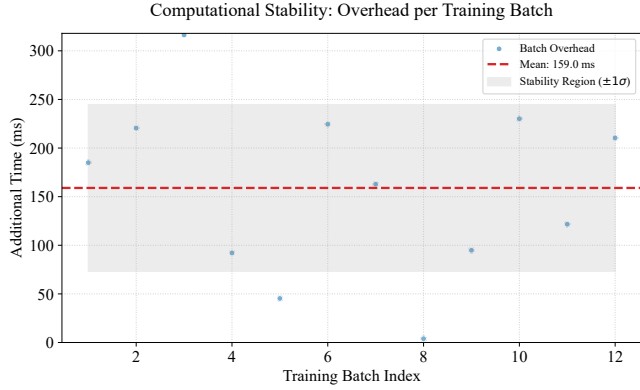

*Figure 9.* **Computational Stability Analysis.** The scatter plot details the overhead introduced by *RieSelect* at each training batch for the T5-Large model. The red dashed line represents the mean overhead ($\approx 159$ ms), and the gray area indicates the stability region ($\pm 1$ standard deviation). The consistent distribution confirms that our geometric operations impose a predictable and stable computational cost.

## D. More Details of Experimental Settings and Results

### D.1. More Details of Datasets

We evaluate on SuperNI and Long Sequence benchmarks. SuperNI comprises 15 diverse NLP tasks (including generation and extraction) organized into two task orders (Order 1/2), while Long Sequence contains 15 classification tasks forming Order 3/4. For completeness, we list the task composition and the exact task sequences used in our experiments in Tables 3–5.

### D.2. More Implementation Details

We implement all methods with the same instruction-tuning pipeline using LoRA adapters on two backbones, T5-Large and LLaMA2-7B. All experiments were conducted with 8 NVIDIA RTX A6000 GPUs (48GB). For LLaMA training, we adopt FlashAttention-2 (Dao, 2024) to reduce memory consumption and accelerate attention computation. We use bf16 training and gradient checkpointing for memory efficiency.

We simulate a federated setting with 50 clients and sample 5 clients per global round. For each task, we run 5 global rounds and each selected client performs 10 local training epochs. Client data are partitioned by a Dirichlet split with concentration parameter $\alpha = 10$ to model statistical heterogeneity.

We use per-device batch size 16 with gradient accumulation steps 2. For sequence lengths, we set $max\_source\_length = 512$. For target lengths, we use $max\_target\_length = 512$ for Orders 1–2 (SuperNI, generation-style tasks), and $max\_target\_length = 256$ for Orders 3–4. The learning rates $\eta_0$ are set as follows: for LLaMA, $5 \times 10^{-5}$ on Order 1 and $1 \times 10^{-5}$ on Orders 2–4; for T5-Large, $2 \times 10^{-4}$ on Orders 1–2 and $1 \times 10^{-4}$ on Orders 3–4.

Unless otherwise specified, LoRA rank is fixed to $r = 8$ for all orders. To ensure a fair comparison across architectures, we

*Table 3.* Details of tasks in the Long Sequence benchmark (Order 3/4).

| Dataset name | Category | Domain | Task Type | Metric |
|---|---|---|---|---|
| Yelp | CL Benchmark | sentiment analysis | Yelp reviews | Accuracy |
| Amazon | CL Benchmark | sentiment analysis | Amazon reviews | Accuracy |
| DBpedia | CL Benchmark | topic classification | Wikipedia | Accuracy |
| Yahoo | CL Benchmark | topic classification | Yahoo Q&A | Accuracy |
| AG News | CL Benchmark | topic classification | news | Accuracy |
| MNLI | GLUE | natural language inference | various | Accuracy |
| QQP | GLUE | paraphrase detection | Quora | Accuracy |
| RTE | GLUE | natural language inference | news, Wikipedia | Accuracy |
| SST-2 | GLUE | sentiment analysis | movie reviews | Accuracy |
| WiC | SuperGLUE | word sense disambiguation | lexical databases | Accuracy |
| CB | SuperGLUE | natural language inference | various | Accuracy |
| COPA | SuperGLUE | question and answering | blogs, encyclopedia | Accuracy |
| BoolQA | SuperGLUE | boolean question answering | Wikipedia | Accuracy |
| MultiRC | SuperGLUE | question and answering | various | Accuracy |
| IMDB | SuperGLUE | sentiment analysis | movie reviews | Accuracy |

*Table 4.* Details of tasks in the SuperNI benchmark (Order 1/2).

| Dataset name | Task Type | Metric |
|---|---|---|
| Task639_multi_woz_user_utterance_generation | summarization | Rouge-L |
| Task1590_diplomacy_text_generation | summarization | Rouge-L |
| Task1729_persona_chat_generate_next | summarization | Rouge-L |
| Task181_outcome_extraction | information extraction | Rouge-L |
| Task748_glucose_reverse_cause_event_detection | information extraction | Rouge-L |
| Task1510_evaluation_relation_extraction | information extraction | Rouge-L |
| Task002_quoref_answer_generation | dialogue generation | Rouge-L |
| Task073_commonsenseqa_answer_generation | dialogue generation | Rouge-L |
| Task591_sciq_answer_generation | dialogue generation | Rouge-L |
| Task511_reddit_tifu_long_text_summarization | question answering | Rouge-L |
| Task1290_xsum_summarization | question answering | Rouge-L |
| Task1572_samsum_summary | question answering | Rouge-L |
| Task363_sst2_polarity_classification | sentiment analysis | Accuracy |
| Task875_emotion_classification | sentiment analysis | Accuracy |
| Task1687_sentiment140_classification | sentiment analysis | Accuracy |

apply LoRA adapters specifically to the query ($q$) and value ($v$) projection matrices in all attention layers for both T5-Large and LLaMA2-7B, while keeping other linear modules frozen. For methods that introduce task-specific LoRA modules (e.g., O-LoRA and N-LoRA), we also set the rank of each newly added LoRA module to $r = 8$. For O-LoRA, N-LoRA, and C-LoRA, we set the regularization hyperparameter to $0.3$. For HydraLoRA, we follow the original paper and use 4 experts, with each LoRA expert set to rank $r = 32$. For traditional CL baselines, we set the EWC regularization coefficient to $5000$; for replay, we store 50 samples per task; for A-GEM, we use a buffer of 500 samples to approximate gradients of past tasks and set the threshold hyperparameter to $0.5$. For LoRM, besides LoRA updates, we additionally upload an approximate Gram matrix per layer; for PILoRA, we additionally upload class prototypes. In our method, we normalize the Fisher matrix to prevent gradient explosion and we set $\vartheta = 0.3$ and $\varsigma = 0.3$, and use $R_\epsilon = 10$ by default. In all experiments, we set $\lambda = 1$ by default and use a fixed task-specific EMA decay of $\gamma = 0.3$. We maintain a diagonal Fisher surrogate $F_{\text{curr}}^{(\ell)}$ online during local training. At each mini-batch step, we form $F_{\text{batch}} := g \odot g$ and update $F_{\text{curr}}^{(\ell)} \leftarrow 0.9\, F_{\text{curr}}^{(\ell)} + 0.1\, F_{\text{batch}}^{(\ell)}$.

In budget-matched comparisons (Sec. 5.3), we impose an uplink budget $P$ (MB) for each participating client in each global round, which limits how many layers' updates can be transmitted. We measure communication by payload size using decimal units. Under bf16, each transmitted parameter costs 2 bytes; if layer $\ell$ uploads $n_\ell$ parameters, its cost is $c_\ell = 2n_\ell/10^6$ (MB), and we select an uploaded layer set $\mathcal{Z}$ satisfying $\sum_{\ell \in \mathcal{Z}} c_\ell \leq P$. We report the total uplink volume as Comm. (GB) by aggregating all client uploads across all sampled clients, all global rounds, and all tasks:

$$\text{Comm} = \frac{1}{10^9} \sum_{u=1}^{N_{\text{uplink}}} \text{bytes}_u = \frac{N_{\text{uplink}} \cdot \overline{\text{bytes/uplink}}}{10^9},$$

*Table 5.* Task orders used in our experiments.

| Benchmark | Order | Task Sequence |
|---|---|---|
| SuperNI | 1 | task1572 → task363 → task1290 → task181 → task002 → task1510 → task639 → task1729 → task073 → task1590 → task748 → task511 → task591 → task1687 → task875 |
| SuperNI | 2 | task748 → task073 → task1590 → task639 → task1572 → task1687 → task591 → task363 → task1510 → task1729 → task181 → task511 → task002 → task1290 → task875 |
| LongSeq | 3 | MNLI → CB → WiC → COPA → QQP → BoolQA → RTE → IMDB → Yelp → Amazon → SST-2 → DBpedia → AG News → MultiRC → Yahoo |
| LongSeq | 4 | Yelp → Amazon → MNLI → CB → COPA → QQP → RTE → IMDB → SST-2 → DBpedia → AG News → Yahoo → MultiRC → BoolQA → WiC |

*Table 6.* **Performance on T5-Large.** AA (%) and BWT (%) on SuperNI and Long Sequence. **Comm.** denotes the total *uplink* volume (GB) over all tasks, including LoRA updates and method-specific auxiliary statistics, such as Fisher matrices or prototype metadata. **Bold** and underlined numbers indicate the best and second-best results. Improvement reports gains over the best non-ours baseline.

| Type | Method | Comm. (GB) | SuperNI | | | | LongSeq | | | |
|---|---|---|---|---|---|---|---|---|---|---|
| | | | Order 1 | | Order 2 | | Order 3 | | Order 4 | |
| | | | AA↑ | BWT↑ | AA↑ | BWT↑ | AA↑ | BWT↑ | AA↑ | BWT↑ |
| PEFT-CL | LoRA (Hu et al., 2022) | 1.76 | 16.30 | −26.58 | 26.32 | −17.27 | 18.76 | −64.42 | 6.44 | −78.25 |
| | O-LoRA (Wang et al., 2023) | 1.76 | 21.69 | −8.58 | 27.22 | −6.41 | 29.92 | −35.28 | 16.53 | −40.86 |
| | N-LoRA (Yang et al., 2025a) | 1.76 | 20.52 | −10.81 | 25.24 | −8.00 | 27.33 | −41.66 | 11.31 | −58.84 |
| | HydraLoRA (Tian et al., 2024) | 4.84 | 22.34 | −19.33 | 28.86 | −14.09 | 18.62 | −64.29 | 6.44 | −77.85 |
| | C-LoRA (Smith et al., 2023) | 1.76 | 25.30 | −13.28 | 30.62 | −11.65 | 38.23 | −40.14 | 13.68 | −63.18 |
| CL | EWC (Kirkpatrick et al., 2017) | 3.52 | 20.87 | -7.04 | 22.72 | −6.13 | 40.12 | −14.61 | 55.33 | **-1.50** |
| | Replay (Chaudhry et al., 2019b) | 1.76 | 26.45 | −8.91 | 31.05 | −5.76 | 50.10 | −18.48 | 46.94 | −16.30 |
| | A-GEM (Chaudhry et al., 2019a) | 1.76 | 24.73 | −18.08 | 31.62 | −11.75 | 21.80 | −58.55 | 34.43 | −46.42 |
| FCL | LoRM (Salami et al., 2025) | 2.82 | 23.64 | −12.95 | 27.84 | −12.39 | 34.54 | −43.38 | 11.43 | −70.31 |
| | PILoRA (Guo et al., 2024) | 3.52 | 21.11 | −18.24 | 28.42 | −14.27 | 14.07 | −71.31 | 10.99 | −71.97 |
| Ours | **RieSelect** | **0.09** | **40.28** | **4.31** | **43.35** | **3.51** | **78.24** | **0.70** | **75.46** | -2.49 |
| | Improvement | ↓ 95.0% | +13.83 | +11.35 | +11.73 | +9.27 | +28.14 | +15.31 | +20.13 | -0.99 |

where in our setup $N_{\text{uplink}} = 5 \times 5 \times 15 = 375$. In Sec. 5.3, we set $P$ to match the average per-client per-round payload of RieSelect. Concretely, the reported total uplink volumes imply $\overline{\text{bytes/uplink}} \approx 0.24$ MB on T5-Large (0.09 GB over $N_{\text{uplink}}$=375 uploads) and $\approx 0.19$ MB on LLaMA2-7B (0.07 GB over 375 uploads), i.e., about $\approx 5\%$ and $\approx 2.5\%$ of the corresponding full-LoRA uplink payloads, respectively.

For the communication overhead of baselines involving auxiliary statistics, we follow standard federated protocols. Specifically, for EWC, clients transmit both the updated LoRA parameters and the corresponding diagonal Fisher Information Matrix (which has the same dimensionality as the parameters), resulting in a total uplink volume of $2\times$ the standard LoRA baseline. Similarly, for PILoRA and HydraLoRA, the additional costs (e.g., prototypes or multiple experts) are accounted for based on their respective architectural designs.

### D.3. Additional Results for T5-Large

To keep the main text focused, we report the full T5-Large main-results table in the appendix. Appendix Table 6 reports AA/BWT on SuperNI (Orders 1/2) and Long Sequence (Orders 3/4), together with the total uplink traffic (Comm.), following the communication accounting in Appendix D.2. RieSelect achieves the strongest AA across all orders while sharply reducing uplink (e.g., 0.09 GB vs. 4.84 GB for HydraLoRA on T5-Large), supporting our claim that selective transmission can improve both adaptation and efficiency. The table also illustrates that stability should be assessed jointly with plasticity: methods with less-negative BWT can still underfit new tasks and yield much lower AA (e.g., EWC on LongSeq Order 4).

### D.4. Additional Results for Qwen2.5-14B-Instruct

To further examine whether the effectiveness of RieSelect is confined to the backbones used in the main experiments, we additionally evaluate Qwen2.5-14B-Instruct on SuperNI Order 2 under the same federated continual learning protocol. As shown in Table 7, RieSelect achieves the highest AA with positive BWT, providing additional evidence that its benefit is not

*Table 7.* Additional evaluation on Qwen2.5-14B-Instruct using SuperNI Order 2. Each cell reports AA (%) and BWT (%).

| Method | AA↑ | BWT↑ |
|---|---|---|
| LoRA | 35.09 | -13.25 |
| O-LoRA | 38.33 | -11.19 |
| N-LoRA | 11.49 | 0.06 |
| HydraLoRA | 36.00 | -10.53 |
| C-LoRA | 37.43 | -5.14 |
| EWC | 38.93 | -10.34 |
| Replay | 38.49 | -9.16 |
| A-GEM | 35.69 | -12.86 |
| LoRM | 32.73 | -7.76 |
| PILoRA | 35.20 | -12.82 |
| RieSelect | **52.29** | **0.25** |

*Table 8.* Robustness under stronger Dirichlet heterogeneity on SuperNI Order 2 with LLaMA2-7B. Each cell reports AA / BWT (%). Smaller $\alpha$ indicates stronger non-IID heterogeneity.

| Method | $\alpha = 10$ | $\alpha = 1$ | $\alpha = 0.5$ | $\alpha = 0.1$ |
|---|---|---|---|---|
| LoRA | 31.14 / -15.96 | 38.58 / -9.23 | 39.25 / -10.42 | 36.39 / -13.44 |
| O-LoRA | 37.43 / 0.50 | 36.00 / 0.04 | 36.64 / 0.80 | 36.59 / 0.75 |
| N-LoRA | 35.22 / 0.85 | 35.34 / 1.83 | 35.32 / 1.41 | 34.91 / 0.88 |
| HydraLoRA | 36.96 / -5.78 | 37.52 / -3.59 | 37.63 / -4.83 | 39.08 / -2.91 |
| C-LoRA | 38.11 / -2.05 | 37.87 / -1.59 | 38.14 / -1.35 | 38.41 / -1.58 |
| EWC | 37.94 / 0.31 | 37.72 / -0.09 | 38.26 / 0.32 | 38.84 / 0.36 |
| Replay | 37.67 / -5.14 | 32.75 / 3.19 | 39.44 / -5.82 | 31.92 / **5.90** |
| A-GEM | 40.92 / -1.61 | 40.48 / -1.17 | 41.74 / 0.15 | 37.87 / -3.64 |
| LoRM | 36.82 / -1.50 | 32.65 / -5.87 | 36.75 / -1.44 | 37.21 / -1.85 |
| PILoRA | 37.51 / -5.41 | 36.46 / -5.07 | 37.69 / -4.55 | 40.02 / -2.12 |
| RieSelect | **44.46 / 5.08** | **45.39 / 5.06** | **45.47 / 3.82** | **44.81 / 3.21** |

limited to the T5-Large and LLaMA2-7B settings.

### D.5. Robustness under Stronger Non-IID Heterogeneity

We further evaluate RieSelect under stronger non-IID client data distributions by decreasing the Dirichlet concentration parameter $\alpha$. Smaller $\alpha$ indicates more skewed client data. Table 8 reports results on SuperNI Order 2 with LLaMA2-7B. RieSelect consistently achieves the highest AA across all tested heterogeneity levels and maintains positive BWT, suggesting that its replay-free conflict estimation remains effective under stronger client heterogeneity.

### D.6. Additional Results for Sec. 5.3

Sec. 5.3 shows that under a matched per-round uplink budget, naive sparsification can severely hurt continual adaptation, and the gains of RieSelect mainly come from allocating the limited budget to the *right* layers rather than sparsity itself. Following Fig. 4, we consider two representative budget-matched baselines: (i) **Top-K layer upload**, which greedily uploads layers with the largest $\ell_2$ norms of layer-wise updates until the budget is met; and (ii) **Random layer upload**, which uploads a random subset of layers under the same budget. We report additional results on SuperNI Order 2 and Long Sequence Order 3.

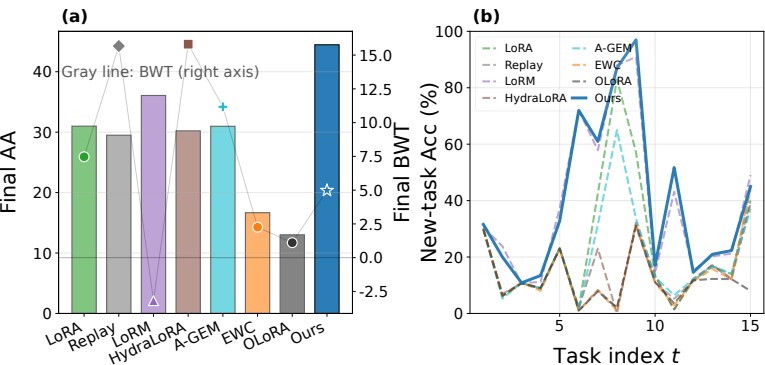

*Figure 10.* Budget-matched **Top-K** layer uploads on SuperNI Order 2. (a) Final AA (left axis) and final BWT (right axis). (b) New-task accuracy along the sequence. Under the same uplink budget, Top-K baselines can lose adaptation on parts of the sequence, whereas RieSelect remains consistently strong.

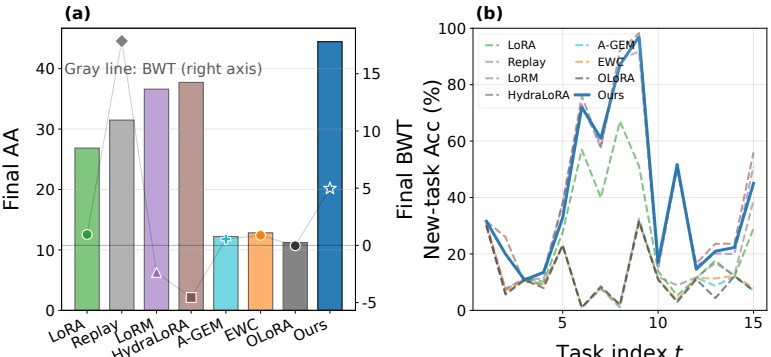

*Figure 11.* Budget-matched **Random** layer uploads on SuperNI Order 2. Randomly spending the same budget does not reliably preserve useful learning signals, leading to weaker overall AA, while RieSelect maintains stronger adaptation.

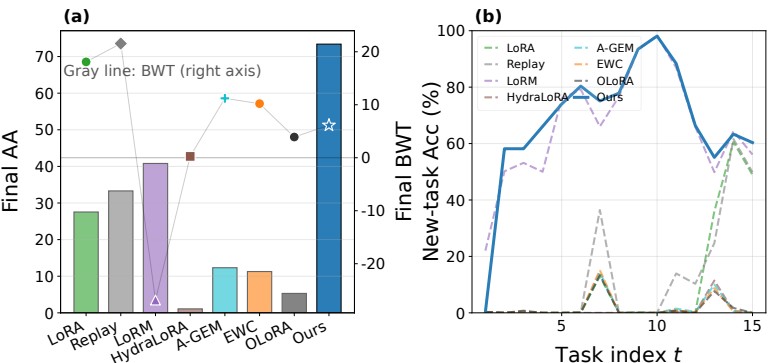

*Figure 12.* Budget-matched **Random** layer uploads on Long Sequence Order 3. Random selection can drastically impair continual learning under tight budgets: new-task accuracy for many baselines collapses for a large portion of the sequence, and any apparent stability may reflect underfitting rather than successful retention. In contrast, RieSelect preserves strong new-task accuracy and achieves the best final AA under the same budget.

