# OpenReview forum: "Less Is More in Federated Continual Learning: RieSelect for Conflict-Aware Layer Selection in LLMs"
_ICML.cc/2026/Conference — ICML 2026 regular_

### Official Review · Reviewer_jpaf · 2026-03-13

**Soundness:** 3
**Presentation:** 3
**Significance:** 2
**Originality:** 3
**Overall Recommendation:** 4
**Confidence:** 3

**Summary:**

This paper addresses the challenges of Federated Continual Learning (FCL) for LLMs deployed on edge devices. It reveals an insightful "less-is-more" phenomenon, demonstrating that cross-task conflicts are highly concentrated in a small subset of layers, and subsequently proposes the RieSelect algorithm. Specifically, RieSelect constructs a Riemannian safe basin using the Fisher Information Matrix to constrain parameter updates. It mitigates catastrophic forgetting by identifying high-conflict layers and applying a closed-form certified safe step size.

**Compliance With Llm Reviewing Policy:**

Affirmed.

**Final Justification:**

My concerns have been addressed, and I will keep my original score.

**Key Questions For Authors:**

See the weaknesses.

**Limitations:**

yes

**Strengths And Weaknesses:**

Strengths
1. Contrary to the conventional wisdom that more communication or parameter updates yield better learning performance, the authors empirically demonstrate that excessive parameter uploading introduces updates from high-conflict layers, which severely disrupt previously acquired knowledge.
2. The introduction of Riemannian geometry and the Fisher Information metric provides a rigorous and elegant mathematical formulation for addressing the catastrophic forgetting problem in continual learning.

Weaknesses
1. The models evaluated in the paper are somewhat outdated and relatively small in scale. Consequently, it remains unclear whether RieSelect can maintain its efficacy and scalability when applied to modern LLM architectures with significantly larger parameter counts.
2. RieSelect conducts conflict assessment and selection at the layer level. However, within a single layer of an LLM, conflicts might be driven by only a small fraction of the weights. Filtering or constraining updates at the macro layer level could inadvertently discard non-conflicting parameter updates that are highly beneficial for learning new tasks.
3. The paper lacks experiments under highly heterogeneous (extreme Non-IID) data distributions, such as Dirichlet distributions with $\alpha=0.1$ or $\alpha=1$. In scenarios with severe data skewness, the locally computed Fisher Information Matrices and gradient conflict directions on clients could deviate significantly, potentially compromising the robustness and effectiveness of the RieSelect algorithm.

---

> ### Author Rebuttal · Authors · 2026-03-30
>
> We thank reviewer *jpaf* for their feedback and careful reading. We address the reviewer's concerns below.
>
> **1. Generalization to newer/larger LLMs.**
>
> Thank you for raising this point. To address it, we additionally evaluate Qwen2.5-14B-Instruct on the same SuperNI Order 2 setting used in the main paper (Table R3-1). RieSelect still achieves the best AA (52.29) and positive BWT (0.25), while all baselines are substantially worse in AA and mostly negative in BWT. This shows that RieSelect is not limited to the original LLaMA2-7B / T5-Large setup and remains effective on a newer, substantially larger LLM backbone. We will add this experiment and Table R3-1 in the revision.
>
> **Table R3-1. Qwen2.5-14B-Instruct on SuperNI Order 2 ($\alpha=10$).**
>
> | Method    |        AA |      BWT |
> | --------- | --------: | -------: |
> | LoRA      |     35.09 |   -13.25 |
> | O-LoRA    |     38.33 |   -11.19 |
> | N-LoRA    |     11.49 |     0.06 |
> | HydraLoRA |     36.00 |   -10.53 |
> | C-LoRA    |     37.43 |    -5.14 |
> | EWC       |     38.93 |   -10.34 |
> | Replay    |     38.49 |    -9.16 |
> | A-GEM     |     35.69 |   -12.86 |
> | LoRM      |     32.73 |    -7.76 |
> | PILoRA    |     35.20 |   -12.82 |
> | RieSelect | **52.29** | **0.25** |
>
> **2. Why layer-level rather than finer-grained conflict control?**
>
> Thank you for raising this point. We agree that conflicts within a layer can in principle be non-uniform, so a finer-grained scheme might preserve some beneficial within-layer updates. However, our layer-level design is motivated by robustness rather than convenience. Appendix B.2 shows that, under weak dependence, aggregating coordinate-level interaction scores into a layer score reduces normalized variance and yields a more stable sign decision, whereas coordinate-wise filtering is more brittle under mini-batch noise and client heterogeneity. This granularity also reduces the candidate space from $d$ coordinates to $L$ layer blocks, making the subsequent budgeted selection tractable with $L \ll d$.
>
> Importantly, a conflicting layer is not automatically discarded. RieSelect follows identify $\rightarrow$ adjust $\rightarrow$ select: conflicting layers are first rectified by the safe-step controller and only then enter budgeted selection. Thus, layer-level control does not simply zero out all updates within a flagged layer. Empirically, Fig. 1(b-d) shows that conflict is concentrated in a small subset of layers, while Fig. 3 shows that RieSelect still maintains strong new-task learning rather than uniformly suppressing updates. We will clarify this robustness–granularity trade-off more explicitly in the revision.
>
> **3. Robustness under extreme non-IID heterogeneity.**
>
> Thank you for raising this point. To address it, we additionally evaluate SuperNI Order 2 with LLaMA2-7B under stronger Dirichlet heterogeneity levels $\alpha \in \{1, 0.5, 0.1\}$, with $\alpha=10$ as reference. The full results are reported in Table R3-2.
>
> Empirically, RieSelect remains robust under these extreme non-IID settings. It achieves the highest AA for all tested $\alpha$ and keeps BWT positive throughout. In particular, even under the extreme non-IID setting $\alpha=0.1$, it still attains 44.81 AA and 3.21 BWT. Compared with the original $\alpha=10$ setting (44.46 AA, 4.98 BWT), the overall performance remains stable rather than collapsing. If severe data skewness substantially corrupted the replay-free Fisher estimate and conflict directions, we would expect a clear degradation in AA or BWT as $\alpha$ decreases, which is not observed here. These results suggest that such extreme non-IID heterogeneity does not materially undermine RieSelect in this setting. We will add this experiment and discussion in the revision.
>
> **Table R3-2. SuperNI Order 2 under different Dirichlet heterogeneity levels (LLaMA2-7B). Each cell reports AA / BWT.**
>
> | Method    | $\alpha=10$      | $\alpha=1$       | $\alpha=0.5$     | $\alpha=0.1$     |
> | --- | --- | --- | --- | --- |
> | LoRA      | 31.14 / -15.96   | 38.58 / -9.23    | 39.25 / -10.42   | 36.39 / -13.44   |
> | O-LoRA    | 37.43 / 0.50     | 36.00 / 0.04     | 36.64 / 0.80     | 36.59 / 0.75     |
> | N-LoRA    | 35.22 / 0.85     | 35.34 / 1.83     | 35.32 / 1.41     | 34.91 / 0.88     |
> | HydraLoRA | 36.96 / -5.78    | 37.52 / -3.59    | 37.63 / -4.83    | 39.08 / -2.91    |
> | C-LoRA    | 38.11 / -2.05    | 37.87 / -1.59    | 38.14 / -1.35    | 38.41 / -1.58    |
> | EWC       | 37.94 / 0.31     | 37.72 / -0.09    | 38.26 / 0.32     | 38.84 / 0.36     |
> | Replay    | 37.67 / -5.14    | 32.75 / 3.19     | 39.44 / -5.82    | 31.92 / **5.90** |
> | A-GEM     | 40.92 / -1.61    | 40.48 / -1.17    | 41.74 / 0.15     | 37.87 / -3.64    |
> | LoRM      | 36.82 / -1.50    | 32.65 / -5.87    | 36.75 / -1.44    | 37.21 / -1.85    |
> | PILoRA    | 37.51 / -5.41    | 36.46 / -5.07    | 37.69 / -4.55    | 40.02 / -2.12    |
> | RieSelect | **44.46 / 4.98** | **45.39 / 5.06** | **45.47 / 3.82** | **44.81** / 3.21 |

---

> > ### Author Rebuttal · Reviewer_jpaf · 2026-04-03
> >
> > The rebuttal solved my concerns, so I maintain my score.

---

> > > ### Author Response · Authors · 2026-04-04
> > >
> > > Dear Reviewer jpaf,
> > >
> > > Thank you for your follow-up and for indicating that our rebuttal has resolved your concerns. We sincerely appreciate your time and consideration.
> > >
> > > Best regards,
> > >
> > > The Authors

---

### Official Review · Reviewer_fqiM · 2026-03-13

**Soundness:** 3
**Presentation:** 3
**Significance:** 3
**Originality:** 3
**Overall Recommendation:** 5
**Confidence:** 4

**Summary:**

This paper addresses the "communication-stability-plasticity trilemma" in Federated Continual Learning (FCL) for LLMs by identifying a "less-is-more" phenomenon where dense parameter uploads exacerbate gradient conflicts and forgetting. The authors propose RieSelect, a three-stage framework that uses Fisher-based conflict detection to identify high-risk layers, applies a Riemannian safe-basin constraint with a closed-form certified safe step size to rectify updates, and employs a 0-1 knapsack formulation to select high-utility, low-risk layers for transmission under a strict uplink budget. Experiments show RieSelect achieves the best average accuracy across task orders.

**Compliance With Llm Reviewing Policy:**

Affirmed.

**Final Justification:**

My concerns  have been addressed, I consider to raise my score accordingly.

**Key Questions For Authors:**

See weakness.

**Limitations:**

To achieve replay-free continual learning, RieSelect requires each client to locally maintain and update two sets of auxiliary Fisher statistics, $\bar{F}$ and $\bar{z}$. Since these tensors have the same dimensionality as the trainable LoRA parameters, they effectively double the local storage and memory footprint required for the adaptation process.

**Strengths And Weaknesses:**

Strength:

1. The motivation is good. The discovery that cross-task conflict is heavy-tailed and concentrated in a few layers provides a strong motivation for selective sparse communication.
2. The use of Riemannian geometry and curvature comparison theorems to derive a certified safe step size moves beyond simple heuristic pruning.
3. The experiment results show RieSelect demonstrates massive reductions in uplink traffic  while significantly improving accuracy over state-of-the-art baselines.
4. The method is replay-free and operates under a fixed per-round budget, making it ideal for privacy-sensitive edge deployments.

Weaknesses:

1. The experiment results show RieSelect demonstrates massive reductions in uplink traffic, but there’s no report how many lora upload to the server for each client. That is how to calculate the uplink traffic for RieSelect?
2. In Eq.10, authors give the adaptive learning rate for RieSelect, can authors show the result of learning rate in each client on each task? And how to calculate $\eta_{safe}$ in real experiment?
3. In experiments, authors use $\alpha=10$ for Dirichlet split, which represents mild heterogeneity even IID. How does RieSelect perform when $\alpha$ is decreased to 0.5 or 0.1? Specifically, does the estimation error of the replay-free Fisher matrix at the edge significantly degrade the safe basin calculation?
4. Table 1 and Table 6 report positive BWT for RieSelect in multiple orders and all the other methods are negative. Does this indicate true Forward Transfer where task A helps task B, or is it simply because the model learns so little from new tasks (due to step-size capping) that the performance on old tasks remains static?

---

> ### Author Rebuttal · Authors · 2026-03-30
>
> We thank reviewer *fqiM* for their feedback and careful reading. We address the reviewer's concerns below.
>
> **1. LoRA upload count and uplink accounting**
>
> Thank you for raising this point. Appendix D.2 defines the communication accounting, though we agree it should be stated more clearly in the main text. For RieSelect, uplink traffic is the total payload of transmitted trainable LoRA layers over selected client-rounds. Because the uploaded subset is dynamic under a fixed per-round budget, uploaded layer counts vary across client-rounds, so we report payload rather than a fixed count. Replay-free Fisher summaries are local and not uploaded. Under the default setup in Appendix D.2, this corresponds to about 0.19 MB per selected client-round on LLaMA2-7B (about 2.5\% of full LoRA transmission).
>
> **2. Client/task learning-rate statistics and $\eta_{\text{safe}}$**
>
> Thank you for raising this point. RieSelect does not use a single adaptive learning rate per client. For conflicting layers, $\eta_{\text{safe}}$ is computed online from Eq. (8) using $S_\ell$, $Q_\ell$, and $\hat{\beta}_\ell$, then clipped by the base learning rate. For non-conflicting layers, Eq. (10) is used. Thus the applied step size is layer-wise.
>
> We additionally report Table R2-1 for three sampled clients (13, 39, 42) at tasks 4, 9, and 14. Here, $\eta$ is the mean applied layer-wise step size, and Cap\% / Conf\% denote capped / conflicting layer fractions. $\eta$ stays close to the base learning rate, while Cap\% is much smaller than Conf\%, showing selective rather than uniform capping. We will add the full table in the revision.
>
> **Table R2-1. Ranges of adaptive statistics across sampled clients.**
>
> | Task | $\eta$ ($\times 10^{-5}$) | Cap\%       | Conf\%       |
> | ---- | ------------------------- | ----------- | ------------ |
> | 4    | 0.88--0.96                | 4.30--15.23 | 63.67--85.94 |
> | 9    | 0.98--0.99                | 1.95--2.34  | 66.80--85.55 |
> | 14   | 0.98--1.00                | 0.00--2.34  | 90.23--99.22 |
>
> **3. Robustness under stronger heterogeneity**
>
> Thank you for raising this point. We agree that $\alpha=10$ is relatively mild. To address this concern, we additionally evaluate SuperNI Order 2 with LLaMA2-7B under $\alpha=1, 0.5, 0.1$, with $\alpha=10$ as reference. For brevity, the full comparison is reported in Reviewer 3’s third question (Table R3-2), which raises essentially the same heterogeneity issue.
>
> Empirically, RieSelect remains robust under substantially stronger non-IID settings. For RieSelect, AA remains stable (44.46 at $\alpha=10$, 45.47 at $\alpha=0.5$, and 44.81 at $\alpha=0.1$), while BWT stays positive (4.98, 3.82, and 3.21). RieSelect remains best in AA across all tested $\alpha$ and keeps positive BWT throughout. These results suggest that the replay-free Fisher estimate on clients remains reliable for the safe-basin calculation in this setting even at $\alpha=0.5$ and $0.1$, and we will add this experiment and discussion in the revision.
>
> **4. Is positive BWT due to under-learning?**
>
> Thank you for raising this question. Table 1/Table 6 report positive BWT (Backward Transfer), not forward transfer, as later learning improves earlier tasks.
>
> We do not think RieSelect’s positive BWT can be simply attributed to weak new-task learning. Rather, it reflects a better stability–plasticity trade-off. To support this point, we additionally report Table R2-2 averaged over the four reported orders. For brevity, Table R2-2 shows representative baselines only, but the same conclusion holds across all methods. RieSelect maintains competitive new-task accuracy while achieving the strongest retained old-task accuracy and the only positive mean BWT on both backbones. This is also consistent with Fig. 3 in the main manuscript, and we will add the full version of Table R2-2 in the revision.
>
> **Table R2-2. Mean new / retained / BWT (A/B: LLaMA2-7B / T5-Large).**
>
> | Method    | New (A/B)         | Retained (A/B)    | BWT (A/B)       |
> | --------- | ----------------- | ----------------- | --------------- |
> | LoRA      | **57.62 / 60.48** | 35.18 / 13.57     | -22.01 / -46.63 |
> | C-LoRA    | 53.71 / 56.88     | 41.79 / 25.01     | -11.65 / -32.06 |
> | Replay    | 45.51 / 50.25     | 40.55 / 38.28     | -5.20 / -12.44  |
> | LoRM      | 53.68 / 56.80     | 39.48 / 22.06     | -13.96 / -34.76 |
> | RieSelect | 55.62 / 57.42     | **58.43 / 59.34** | **2.92 / 2.05** |
>
> **5. Local storage overhead of $\bar{F}$ and $\bar{z}$**
>
> Thank you for raising this point. We agree that this introduces a real local storage trade-off. More precisely, it does not double the full adaptation memory footprint, but adds roughly two extra LoRA-sized states on top of the trainable LoRA parameters. In our LLaMA2-7B setting, the trainable LoRA state is only 4,194,304 out of 6.74B parameters (0.062\%), so the absolute overhead is modest, but this is still a real limitation that we will state more explicitly in the revision.

---

> > ### Author Rebuttal · Reviewer_fqiM · 2026-04-03
> >
> > Thank you for the response, my concerns  have been addressed, I consider to raise my score accordingly.

---

> > > ### Author Response · Authors · 2026-04-04
> > >
> > > Dear Reviewer fqiM,
> > >
> > > Thank you for your follow-up and for your encouraging feedback. We are grateful that our rebuttal has addressed your concerns, and we sincerely appreciate your time and consideration.
> > >
> > > Best regards,
> > >
> > > The Authors

---

### Official Review · Reviewer_tdvk · 2026-03-13

**Soundness:** 3
**Presentation:** 3
**Significance:** 3
**Originality:** 3
**Overall Recommendation:** 4
**Confidence:** 5

**Summary:**

In this paper the author tried to solve the core issue of federated continual learning in llm i.e. communication-stability-plasticity trilemma. Introduced an approach RieSelect  and discovered a phenomenon “less is more” where sending too much data back to the server degrades the performance because some high-conflict layers forget old learned knowledge.

This approach is divided into three steps: identify, adjust, select.
In the first step conflict score is calculated by using the fisher metric layers within the safe score are capped with light basin, others are capped by closed form certificate to stay within the safe basin (Riemannian), capping is done in second step then in third step primary purpose is to select layer which has to be updated in server with constrained budget is formulated as 0-1 knapsack problem.

Several metrics are calculated in edge based on that knapsack is applied like gain,how much forget,conflict penalty,cost etc.

Got 115x data uplink reduction for llama7b and 53x for T5-large compared to SOTA (PiLORA, hydraLORA). Accuracy increased by 18.99 points and  28.14 on LLAMA AND T5 large, consistent good performance across different tasks.

**Compliance With Llm Reviewing Policy:**

Affirmed.

**Key Questions For Authors:**

Q1. Does the value of the safe basin radius vary when shifting different task domains?
Q2. Is there any impact or increase in the computation complexity at the client side in your approach? Please quantify.
Q3. Will this approach work for deep learning architectures? Smaller LLMs? A discussion on generalization for different model types/categories will be useful.

**Limitations:**

Limitation section missing. Kindly write.

**Strengths And Weaknesses:**

Originality: Idea moves beyond simple gradient projection to isolation heavy strategies
Good Results: 18.99 and 28.14 points improvement over SOTA and significant uplink reduce 53x,115x times


Weakness:
Limited diversity in llm architecture

---

> ### Author Rebuttal · Authors · 2026-03-30
>
> We thank reviewer *tdvk* for their feedback and careful reading. We address the reviewer's concerns below.
>
> **1. Does the value of the safe basin radius vary when shifting different task domains?**
>
> Thank you for raising this point. No. In the current method we do not vary $R_\epsilon$ across task/domain shifts. We use a fixed $R_\epsilon=10$ for all task boundaries. This is intentional in strict replay-free FCL, where reliable online estimation of shift magnitude is difficult. Although $R_\epsilon$ is fixed, the induced constraint is not a uniform Euclidean tolerance. Since the safe basin is defined in Fisher geometry, the same radius can impose a stricter constraint in directions associated with higher conflict and a looser one in more compatible directions.
>
> Empirically, Fig. 5 in Sec. 5.4 of the main manuscript shows a clear sweet spot: $R_\epsilon=10$ achieves the best final AA and is the only tested value with positive final BWT on LLaMA2-7B Order 2. To show this is not merely aggregate, we add Table R1, where “Trans. best” counts transitions with the best $AA_k$ and “Final best” counts tasks whose final accuracy is best or tied-best across radii. Table R1 shows that $R_\epsilon=10$ is best on 13/14 transitions and best or tied-best on 11/15 final tasks. These results suggest that although the best radius may vary across shifts, a single moderate radius is a stable practical choice in our setting, and we will add this table in the revision.
>
> **Table R1. Task-wise summary of radius sensitivity.**
>
> | $R_\epsilon$ | Trans. best | Final best | Final BWT |
> | :----------: | :---------: | :--------: | :-------: |
> |      1       |    0/14     |    2/15    |   -2.91   |
> |      5       |    0/14     |    1/15    |   -3.62   |
> |      10      |  **13/14**  | **11/15**  | **+4.98** |
> |      20      |    0/14     |    0/15    |   -6.59   |
> |      50      |    1/14     |    2/15    |   -7.87   |
>
> **2. Is there any impact or increase in the computation complexity at the client side in your approach? Please quantify.**
>
> Thank you for raising this point. Yes, RieSelect slightly increases client-side computation, but the overhead is lightweight both theoretically and empirically. As stated in Sec. 4.4 (last paragraph of the main manuscript), the extra per-round overhead is $O(d)$, where $d$ denotes the number of trainable parameters, because the geometric quantities are computed by element-wise operations with diagonal Fisher surrogates. The layer subset selection further adds only $O(LP)$ via dynamic programming, which is negligible in our setting with $L \ll d$ and fixed uplink budget $P$.
>
> We further quantify this overhead in Appendix C. Fig. 8 shows that the added latency per training batch stays in the millisecond range even as model size grows (e.g., 165.2 ms $\rightarrow$ 184.5 ms for $r=16$ when moving from T5-Base to T5-Large). In the largest evaluated setting (T5-Large, $r=32$), the extra cost is 226.3 ms per batch, compared with a baseline training time of 1400.9 ms, i.e., 16.15% additional overhead. Fig. 9 shows stable overhead across iterations (mean $\approx$ 159 ms), with no obvious latency spikes. Overall, the increase is modest in practice.
>
> **3. Will this approach work for deep learning architectures? Smaller LLMs? A discussion on generalization for different model types/categories will be useful.**
>
> Thank you for raising this point. For smaller LLMs, yes to a limited extent; for broader deep learning architectures, not yet empirically in full generality. In our current paper, we evaluate RieSelect on T5-Large and LLaMA2-7B, covering both encoder-decoder and decoder-only architectures. Thus, our current evidence already supports generalization across multiple PEFT-based transformer LLM settings rather than a single backbone.
>
> More specifically, T5-Large provides evidence beyond the original LLaMA2-7B setup, and Appendix Table 6 reports the T5-Large results under the same protocol. Since RieSelect is built on replay-free gradients, diagonal Fisher surrogates, and layer-wise safe-basin control, its mechanism is not tied to one specific transformer backbone. However, we do not claim empirical validation for all deep learning architectures at this stage. We will clarify this scope explicitly in the revision and state that broader validation beyond PEFT-based transformer models is left for future work.
>
> **4. Limitation section missing. Kindly write.**
>
> Thank you for raising this point. We agree and will add an explicit Limitations and Future Work paragraph in the revision. Specifically, we will clarify that the current study is limited to replay-free FCL with PEFT-based transformer LLMs, and that our empirical validation does not yet extend beyond this setting. We will also state that RieSelect uses a fixed safe-basin radius $R_\epsilon$ rather than an adaptive radius selection strategy. Broader architectural validation and adaptive radius design are left for future work.

---

> > ### Author Rebuttal · Reviewer_tdvk · 2026-04-03
> >
> > Points addressed, but the scores are unchanged.

---

> > > ### Author Response · Authors · 2026-04-04
> > >
> > > Dear Reviewer tdvk,
> > >
> > > Thank you for your follow-up and for acknowledging that our rebuttal has addressed your concerns. We sincerely appreciate your time and consideration.
> > >
> > > Best regards,
> > >
> > > The Authors

---

### Decision · Program_Chairs · 2026-04-30

**Decision:**

Accept (regular)

**Comment:**

This paper addresses the challenges of Federated Continual Learning (FCL) for LLMs deployed on edge devices by identifying a "less-is-more" phenomenon where dense parameter uploads exacerbate gradient conflicts and forgetting. The main concerns of the reviewers (e.g., tdvk, fqiM, and jpaf) are limited diversity in LLM architecture, robustness under stronger heterogeneity, local storage overhead, and generalization to larger LLMs. All the reviewers pointed out that their concerns have been addressed in the rebuttal. Therefore, I am recommending acceptance.